# Dawn and Dusk of Late Cretaceous Basin Inversion in Central Europe

Thomas Voigt [1], Jonas Kley [2], Silke Voigt [3]

[1] Institut für Geowissenschaften, Friedrich-Schiller-Universität Jena, Burgweg 11, 07749 Jena, Germany

[2] Georg-August-Universität Göttingen, Geowissenschaftliches Zentrum, Goldschmidtstraße 3, 37077 Göttingen

[3] Goethe-Universität Frankfurt, Institut für Geowissenschaften, Altenhöferallee 1, 60438 Frankfurt

*Correspondence to*: Thomas Voigt, (voigt@geo.uni-jena.de)

**Abstract.** Central and Western Europe was affected by a compressional tectonic event in the Late Cretaceous, caused by the convergence of Iberia and Europe. Basement uplifts, inverted graben structures and newly formed marginal troughs are the main expressions of crustal shortening. Although the maximum activity occurred in a short period between 90 and 75 Ma, the exact timing of this event is still unclear. Dating of start and end of Late Cretaceous basin inversion gives very different results depending on the applied method. On the basis of borehole data, facies and thickness maps, the timing of basin re-organisation was reconstructed for several basins in Central Europe. The obtained data point to a synchronous start of basin inversion at 95 Ma (Cenomanian), 5 million years earlier than commonly assumed. The end of the Late Cretaceous compressional event is difficult to pinpoint in Central Europe, because regional uplift and salt migration disturb the signal of shifting marginal troughs. Late Campanian to Paleogene strata deposited unconformably on inverted structures indicate slowly declining uplift rates during the latest Cretaceous. The differentiation of separate Paleogene inversion phases in Central Europe does not appear possible at present.

## 1.     Introduction

During the Late Cretaceous, Europe was affected by a compressional event, which led to the deformation of the Central European Basin. This compression induced significant shortening of the basement, accompanied by the uplift of basement anticlines within the basin, inversion of normal faults, which were appropriately oriented to the newly established stress field, and folding of sedimentary pre-inversion sequences above the thick Permian Zechstein salt (e.g., Ziegler, 1987; Baldschuhn et al., 2001; Kockel, 2003; de Jager, 2007; Krzywiec, 2006, 2012; Kley and Voigt, 2008; Kley, 2018). Transpression occurred at normal faults oblique to compression (Deckers and van der Voet 2018; van der Voet et al., 2019). Inverted graben fills and uplifted basement units were eroded and redeposited in newly formed flexural basins (marginal troughs), filled with Upper Cretaceous redeposited syntectonic clastic sediments and/or hemipelagic to pelagic limestones. Late Cretaceous compressive deformation occurred in a belt along the margin of the East European Platform (Fig. 1), and at intraplate structures from southern England across the North Sea and Central Europe up to the basement of the Molasse Basin in front of the Alps, beneath the Alpine nappes on the Helvetian shelf, and to southern France. Their orientation is oblique to the deformation front of the Alpine Orogen.

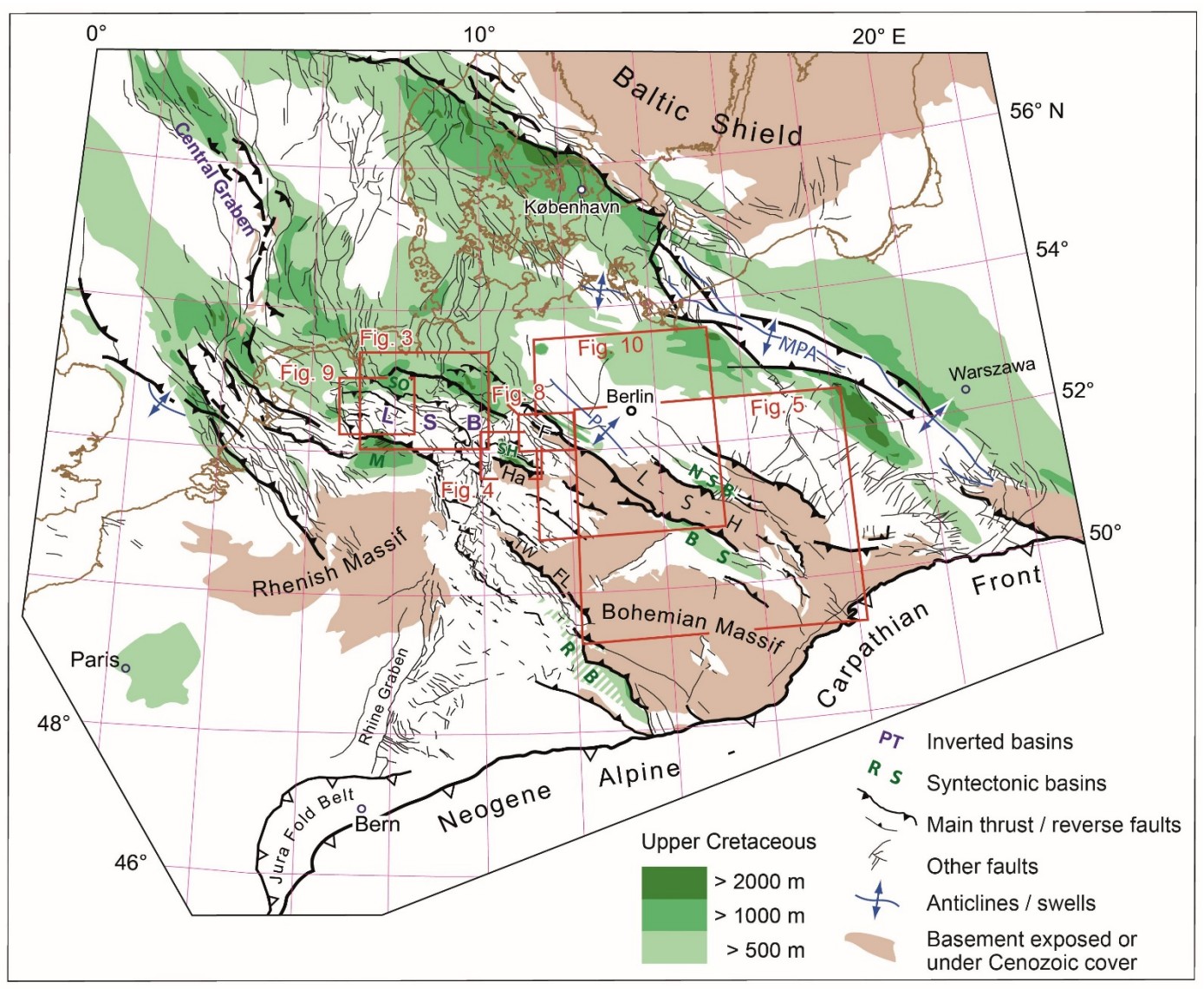

**Fig. 1: Overview of Mesozoic-Cenozoic structures and Late Cretaceous basins in Central Europe and surrounding areas, modified from Kley and Voigt (2008). Cretaceous isopachs are modified from Ziegler (1990). Hatched area in the Regensburg Basin has preserved thickness < 500 m. Red boxes show locations of Figs. 3 to 5 and 8 to 10 as indicated. Abbreviations: LSB Lower Saxony Basin, MPA Mid-Polish Anticlinorium, SO South Oldenburg Basin, M Münsterland Basin, SH Subhercynian Basin, AM Altmark Basin, NSB North Sudetic Basin, BS Bohemian-Saxonian Basin, RB Regensburg Basin, Ha Harz, F Flechtingen High, TW Thüringer Wald, FL Franconian Line, P Prignitz High, L-S-H -Lusatian-Sudetic High.**

## 2. Late Cretaceous Central European Basin deformation – facts and assumptions

The strongest deformation of the European lithosphere is focused on a 200 km wide belt which trends in NW-SE direction and contains numerous basement highs uplifted by several kilometres. It comprises parts of the inverted Lower Saxony Basin, the Harz Mountains, Flechtingen High, the Thuringian Forest with its southern prolongations in Bavaria and the Lusatian-Sudetic High (Senglaub et al. 2006, von Eynatten et al. 2019, Thomson & Zeh, 2000, Hejl et al., 1997, Lange et al., 2008, Danišík et al., 2010). Less pronounced inversion (uplift magnitudes of 500-2000 m) is observed along the margin of the East European platform (Mid-Polish Anticlinorium; e.g. Dadlez, 1980; Krzywiec 2002, 2006; Hansen and Nielsen, 2003; van Buchem et al., 2017), some anticlinal structures of the North German Basin (Prignitz High: Voigt, 2009, Malz, 2019) and Northwestern Europe (Zijerveld et al., 1992; Geluk et al., 1994; Michon et al., 2003; de Jager, 2003; Luijendijk et al., 2011). The amount of vertical displacement may exceed 10 km like in the case of the inversion structures Lower Saxony Basin – Münsterland Basin (Petmecky et al., 1999; Senglaub et al., 2005, 2006), Harz – Subhercynian Basin (von Eynatten et al., 2019) and the Lusatian-Sudetic High – Bohemian-Saxonian Cretaceous Basin (Danišík et al., 2010; Käßner et al., 2020).

Major discussions concern kinematics of deformation. While some authors argued for a NW-SE-directed dextral strike-slip fault system by attributing the uplifts to restraining bends (Ziegler, 1990; Wrede, 1988; Uličný, 2001) and related basins to transtension, most authors agreed that frontal thrusting was the main process to develop the observed structures (Franzke et al., 2004; Kley and Voigt, 2008; Nielsen and Hansen, 2000; Deckers and van der Voet, 2018). This was also confirmed by small-scale structural features (slickensides, fold axes and fault orientations), which in many cases preserved both the extensional phase and N-S to NE-SW convergence (Vandycke, 2002; Franzke et al., 2004; Kley, 2018; Malz et al., 2019; Coubal et al., 2014; Navabpour et al., 2017). The strike-slip model adresses the problem that the principal faults should be orientated in E-W-direction to explain the subsidence anomalies at the assumed releasing Riedel shears, which were in fact never observed. Furthermore, the symmetric shape of the marginal troughs and their spatial relations to the inverted structures strongly point to frontal convergence as the driving force for basin formation (Voigt et al., 2009).

On the basis of a detailed structural analysis of faults, Navabpour et al. (2017) were able to detect an early phase of N-S compression, oblique to the main NW-SE-striking faults, between the extension phase and the frontal thrusting. Nevertheless, this event is not precisely dated yet.

While earlier interpretations emphasized the role of collision in the Alps as the cause of compression (e.g., Ziegler, 1987; Ziegler et al., 2002) or sought the cause for basin inversion in upper mantle processes (Kockel et al., 2003), later authors interpreted the deformation as the result of a general Africa-Europe convergence during the late Cretaceous (Nielsen et al., 2007; Deckers, 2015). Kley and Voigt (2008) emphasized that concurrent deformation occurred in a broad belt from northern Africa (Morocco), and Iberia, across the North Sea and southern England to the Baltic Sea and Poland. It was related to a change in relative motion of the European and African plates, resulting in a short-term Iberia-Europe convergence. With respect to direction or timing, this synchronous compression is not related to any deformation phase in

the Alps. Instead, the opening of the South Atlantic Ocean caused a northward drift of the African Plate and led to a transfer of compression via the Iberian Peninsula to the European Craton and its foreland.

E. Voigt (1963) first recognized the formation of Late Cretaceous "marginal troughs" or thrust-load basins (compare Nielsen and Hansen, 2000; S. Voigt et al., 2008; Hindle and Kley, this issue) in Central Europe, and found their development to be frequently related to the inversion of former basin structures. Nielsen and Hansen (2000) explained the formation of primary marginal troughs through loading by thickened lithosphere of the inverted structures. Primary Late Cretaceous to early Paleocene and secondary late Paleocene marginal troughs, found at the margin of inverted Danish basins, differ in structure and origin (Nielsen et al., 2005). While the former developed due to the load of thickened lithosphere and sediment deposits on the foreland, the latter show a shift of the basin axis away from the inverted structure, and are shallower and wider than the narrow primary basins. They were explained as having evolved due to relaxation of the lithosphere and taken as a marker for a sudden end of inversion tectonics within the early Paleocene (Nielsen et al., 2005; Nielsen et al., 2007). This is in good agreement with the results of Deckers and van der Voet (2018) for the timing of inversion in the Roer Valley Graben and the West-Netherland Basin, respectively. Krzywiec and Stachowska (2016) challenged this concept by emphasizing that the Upper Cretaceous thickness maxima do not represent narrow primary marginal troughs but are due to more complete preservation in synclines. Their example at the southern flank of the Mid-Polish Swell shows a remarkable hiatus below an unconformity overlain by Eocene post-inversion deposits. The lack of Paleocene deposits and pre-Eocene erosion of both marginal troughs and uplifted structures preclude a comparison of original Late Cretaceous vs. Paleogene basin geometries. It is therefore not clear to date whether Palaeocene secondary troughs are restricted to the Danish Basin or had been a common feature of late Cretaceous basin inversion.

A major discussion concerns the continuous or discontinuous nature of deformation during the Late Cretaceous (Subhercynian) inversion (Stille, 1924; Mortimore et al., 1998; T. Voigt et al., 2004; Kley, 2018; Deckers and van der Voet, 2018). In the type region, the Subhercynian Basin at the northern margin of the Harz Mountains, tilted Triassic-Jurassic sedimentary successions are overlain by Upper Cretaceous clastic deposits of different age (summary in T. Voigt et al., 2004). These unconformities span the period from middle Coniacian to lower Campanian (e.g. E. Voigt 1929, Mortimore et al. 1998) and were initially used to distinguish several phases of tilting, erosion and deposition on the newly created erosion surfaces: the Ilsede phase in the Coniacian, the Wernigerode phase (with several sub-phases) in the Santonian/early Campanian and the Peine phase in the late Campanian. Stille (1924) interpreted these phases as separate (and worldwide) tectonic pulses. Following the same line of reasoning, the Laramide phase was imported from Northern America to explain the major unconformity of Eocene deposits overlying Mesozoic and Palaeozoic basement and deformed Permian to Late Cretaceous deposits units through Western and Central Europe (Stille, 1924). Even younger tilting and erosional unconformities were also observed in western sub-basins of the Southern Permian Basin and related to late Eocene ("Pyrenean") and late Oligocene ("Savian") phases of inversion (de Jager, 2007; Deckers et al., 2016).

Mortimore et al. (1998) for the Cretaceous and de Jager (2007) for the Paleogene, correlated these pulses across Western and Central Europe. Mostly, the ages of these unconformities are poorly defined, because they were determined from

sedimentary units covering tilted older rocks. As deposition on such unconformities needs a base-level rise, a single "phase" is often related to major transgressions. The observed five particular late Cretaceous "phases" of the Subhercynian Basin

reflect only the interplay of continuous deformation and changes in base level, which led to phases of erosion and phases of deposition at the margins of continuously active structures. They represent progressive unconformities (T. Voigt et al., 2004). Late Cretaceous marginal troughs within the Polish Basin and the Danish Basin show continuous deformation, expressed by growth strata (Nielsen and Hansen, 2000; Krzywiec, 2002), while unconformities are limited to the margins and tops of inverted structures. More recently, van der Molen et al. (2005), Deckers and van der Voet (2018) and van der

Voet (2019) argued for discrete pulses of inversion in the Netherlands offshore areas. But, as in the case of the Subhercynian Basin, the age of unconformities within the chalk in the southern North Sea seem to correlate to sea-level drops, followed by pronounced transgression (Hancock, 1989). There is no evidence for changing basin configuration during the late Cretaceous inversion; except at the structures oriented oblique to compression (e.g., Dutch Central Graben; van der Voet et al., 2019). The Paleogene inversion is mainly expressed in the western part of the Southern Permian Basin, in the Dutch North Sea area

and western Europe, spanning the middle Paleocene (Laramide phase), the late Eocene (Pyrenean phase; de Jager, 2003; Deckers et al., 2016), and the latest Oligocene/earliest Miocene (Savian phase; de Jager, 2003).

A crude timing of Cretaceous deformation was already established by Ewald (1862) who observed unconformities at the northern margin of one of the most prominent basement structures within the Central European Basin, the Harz Mountains, and concluded a Late Cretaceous age of uplift. Suggestions aimed at a more precise timing were based on several methods,

but came to very different conclusions. Most authors agreed that rapid inversion in central Europe started about 88 million years ago (Coniacian), expressed by rapidly increasing sedimentation rates and a transition from hemipelagic limestones to marly sediments (e.g., Arnold, 1964; Mortimore et al., 1998; Voigt et al., 2006). First evidence of units redeposited by submarine sliding (E. Voigt, 1962) and considerably enhanced thickness of Turonian deposits were taken as markers for the first weak phase of inversion (T. Voigt et al., 2006; Niebuhr et al. 2011, Janetschke and Wilmsen, 2014). Van der Molen et

al (2005) and van der Voet (2019) argued for a late, Santonian or end Campanian start of Subhercynian inversion in the Dutch North Sea.

The fastest uplift of inverted structures and most pronounced subsidence of marginal troughs occurred from Coniacian to Campanian, as reflected by both cooling ages and sedimentation rates. The end of Central European Basin inversion is still under discussion: Intervals proposed by different authors reach from Late Campanian to Danian (70-64 Ma) to even Eocene

or Oligocene (40 Ma). The studies of Deckers (2015), Deckers et al. (2016) and Deckers and van der Voet (2018) showed gentle middle Paleocene undulations of 100-200 km wavelength in and around the southern North Sea, matching a lithospheric folding mechanism but distinct from the Late Cretaceous inversion process. Kley (2018) suggested that Paleogene inversion and uplift in western and Central Europe was unrelated to compression altogether, making it different from Late Cretaceous and younger Cenozoic events concerning both spatial extent and underlying causes. In this paper we

will concentrate on a more precise timing of the Late Cretaceous inversion in Central Europe. Our focus is on basins from the Lower Saxony Basin to the Bohemian-Saxon basin (Fig. 1), with some remarks on regions to the west and east. A second

problem that we want to address is the question whether basin inversion occurred contemporaneously across the whole basin or by successive activation of different fault zones. We present sedimentological data from different marginal troughs of basins in Germany, which pinpoint start and end of basin inversion more precisely. The database was mainly compiled from published isopach maps, thermochronological and seismic data, and the interpretation of sedimentological and geophysical data obtained from cores and boreholes stored at the Geological Surveys of the German federal states of Saxony, Saxony-Anhalt and Lower Saxony.

## 3. Timing of inversion in the basins studied

Investigation of geometrical patterns, in particular seismic stratigraphy of strata deposited during the Jurassic to Early Cretaceous extensional phase in adjacent marginal troughs (e.g., Baldschuhn et al., 1991; Krzywiec, 2006; Nielsen and Hansen, 2000; Vejbæk and Andersen, 2002), thermochronological data from uplifted basement blocks (Hejl et al., 1997; Thomson and Zeh, 2000; Fischer et al., 2012; Lange et al., 2008; Käßner et al., 2020; von Eynatten et al., 2019; Danišík et al., 2010, 2012; Botor et al., 2019) and thermal maturity of the exhumed basin fill (Petmecky et al., 1999; Senglaub et al., 2005, 2006; Luijendijk et al., 2011; Beyer et al., 2014) allowed to constrain the basin history for the majority of active structures. Additionally, the sediment composition (clasts, heavy minerals, and zircon ages) in marginal troughs reflects the rocks that were eroded and redeposited from the uplifting structures and constrains timing and rates of inversion. This method was applied to few basins only, like the Subhercynian Basin (T. Voigt et al., 2006; von Eynatten et al., 2009) and the Bohemian Cretaceous Basin (T. Voigt et al., 2009; Hofmann et al., 2018; Nádaskay et al., 2019; Niebuhr et al. 2020).

According to these data, most authors agree that inversion did not commence before the Late Cretaceous and peaked during Coniacian, Santonian and Campanian times. However, the precise start and end of basin inversion are still debated, according to the variable sensitivity and precision of the applied methods.

## 3.1 Fission track and AHe dating

Low-temperature thermochronology, in particular apatite fission track dating (AFT), has been applied to basement rocks across central Europe (e.g., Hejl et al., 1997; Ventura and Lisker, 2003; Lange et al., 2008; Thomson et al., 1997; Thomson and Zeh, 2000; von Eynatten et al., 2019; Käßner et al., 2020, Danišík et al., 2010, 2012; Botor et al., 2019). The data show in many places a rather homogenous signal of rapid uplift and associated cooling of basement rocks between 90 and 70 Ma (Turonian to Maastrichtian), in some cases continuing to 55 Ma (Palaeocene; von Eynatten et al., 2019; Botor et al., 2019; Sobczyk et al., 2019). Cooling ages from the eastern Sudetes show that the basement of Cretaceous basins underwent a full thermal reset of the AFT-system and was subsequently disrupted (63-45 Ma) by intrabasinal uplifts (Danišík et al. 2012, Sobczyk et al., 2019).

Complete annealing of the apatite fission track system occurs at temperatures above 120-110 °C. Partial annealing with shortening of track lengths on geologically relevant time scales occurs down to about 60 °C. Fully reset samples must have moved rapidly through the ca. 60-50 °C temperature window of the partial annealing zone (PAZ). Estimates of the heat flow during the Cretaceous and results of thermal modelling suggest that the PAZ was about 1.4-2.2 km thick and exhumation of this magnitude is required to cool a sample through the PAZ. Exhumation rates were estimated for the well-constrained case study of the Harz Mountains. Modelled uplift rates based on different cooling ages are in the order of >0.5 km/Ma and in good agreement with the depositional record in the adjacent basin (von Eynatten et al 2019). Earlier estimates were around 1 km/Ma (von Eynatten et al. 2008). It would take a rock residing at the base of the PAZ between 1.4 and 4.0 Ma to rise to the top of the PAZ where its age becomes fixed. The onset of deformation may thus predate the timing of cooling deduced from AFT data by a few million years. This effect is accounted for when time-temperature histories are modelled, but should be considered for ages from older studies, or when only central ages are used for comparison with other data.

Discrepancies between thermochronologic ages and stratigraphic indicators of inversion are evident for the southern basement highs, which were affected by regional uplift, leading also to the partial erosion of adjacent basins. Both in the Bohemian-Saxonian Cretaceous Basin and the North Sudetic Basin that are related to the uplift of the Lusatian-Sudetic High and in the Regensburg Basin, which forms the marginal trough southeast of the Franconian line (Fig. 1), major parts of the basin fill of the marginal troughs were removed. The remaining successions only reflect early stages of the inversion process, because youngest deposits are of early Coniacian to earliest Santonian age. The main stage of basin inversion as known from the northern marginal troughs, is not preserved, although AFT ages point to rapid exhumation and later maximum redeposition, particularly in the Santonian to Campanian. In contrast to the strongly fault-controlled uplift und subsidence during basin inversion, the following regional uplift affected both the source areas and the marginal troughs to regional exhumation and erosion.

## 3.2     Growth strata and progressive unconformities

The evolution of marginal troughs related to basin inversion is caused by thickened crust, which loads and depresses the foreland (e.g., Hansen and Nielsen, 2003). As long as uplifting structures in the inverted basin remain below the erosion level in the early stages of tectonic activity, the thickness of a particular unit is increased in the marginal trough and reduced on top of the uplifting structure. However, if swells and basins remain below the influence of storms and surface currents, sedimentation derives only from "planktonic rain" of coccoliths and foraminifers, which forms a carpet of uniform thickness and thus obliterates the growing structure to some extent (Hancock, 1989). Lykke-Andersen and Surlyk (2004), Surlyk and Lykke-Andersen (2007), van der Molen (2005) and van der Voet et al. (2019) have shown by interpreting seismic profiles that inversion-controlled changes in sea-floor bathymetry have generated both erosional features and current-induced redeposition in pelagic chalk successions below storm wave base, resulting from different strength of bottom current flows. Continuing growth of a swell to above the erosion level leads to erosion, transport and deposition from the swell into the

basin and to formation of growth strata at the margin of the uplifting structure. As the uplift of most structures has proceeded beyond the erosion level, this early stage is rarely preserved and only the thickened basin fill reflects, probably with some delay due to the early position below the erosion level, the tectonic event. Thickening and growth strata can be detected in seismic sections, provided the thickness difference is high enough (Evans and Hopson, 2000; Lykke-Andersen and Surlyk, 2004; Surlyk and Lykke-Andersen, 2007; van Buchem et al., 2017). The resolution depends on the variability of lithology

and seismic impedance.

    Growth structures and unconformities were observed in Upper Cretaceous seismic sections across Europe (e.g., Mortimore et al., 1998; Vejbæk and Andersen, 2002; Nielsen and Hansen, 2000; Krzywiec, 2006; van Buchem et al., 2017) and used to date inversion. Krzywiec and Stachowska (2016) argued that higher total thickness of Upper Cretaceous strata results from folding and erosional truncation at the margin of the inverted structure and to a much lesser degree from increased

subsidence in a marginal trough. The distinction between these two cases is not straightforward and only possible if thickness trends of single units are detectable (Krzywiec, 2006; Krzywiec et al. 2009, Krzywiec and Stachowka, 2016). If the basin margin is involved in the uplift, unconformities can develop. These structures are significant markers of basin deformation, but they occur only in a few places. In the Subhercynian Basin, mainly at the northern margin of the Harz anticline, progressive unconformities related to basin inversion are exposed at the surface. All of them are rotated or affected

by thrusts, indicating that inversion had not ended by the early Campanian. Precise dating of progressive unconformities is critical, because in most cases a time gap between the youngest deformed and the oldest covering units is observed. At the northern margin of the Harz mountains, the first inversion-related unconformity occurs at the base of the middle Santonian (~85 Ma), overlying Upper Triassic to Turonian deposits. Three succeeding unconformities occur in the upper Santonian, in the lower Campanian, and at the base of the Upper Campanian in the northern part of the basin. Further, an older, middle

Coniacian unconformity is exposed at the northern margin of the basin, and composition and thickness of the basin fill shows clearly that inversion started earlier in the Turonian. This time gap at the main structure is caused by progressive tilting of the basin margin and accompanying erosion of older deposits. Older unconformities, which may have been present at the frontal thrust of the Harz mountains, were eroded during the main inversion phase (Voigt et al., 2004). This situation is sometimes misinterpreted in the sense that the overlying sequence post-dates the deformation event immediately. Van

Buchem et al. (2017) described an unconformity on top of the inverted Danish Central Graben at the base of the Upper Campanian to Maastrichtian chalk and argued that inversion was limited to the early Campanian. Seismic sections show nevertheless two well-expressed marginal troughs (Turonian to Lower Campanian) on both sides of the inverted structure, which is characterised by reduced thickness of these units, thus indicating that compression started earlier and was masked by the high sea level during the Late Cretaceous. The unconformity developed from Turonian to early Campanian and

cannot be used as a marker of a short-term tectonic event.

## 3.3    Facies and provenance

Facies changes may occur even in very early stages of basin inversion because facies is mainly controlled by water depth and source areas. Facies changes are observed in Turonian and Coniacian hemipelagic deposits of northern Germany and southern England (e.g., Mortimore et al., 1998, Wilmsen 2003), characterized by changes in composition, fossil diversity and abundance, colour, and occurrence of hardgrounds or condensed sections. These features are mainly caused by carbonate productivity and relationship of the sediment surface to the base level (Wilmsen 2003). Tectonic uplift is difficult to distinguish from processes related to climate change, active salt diapirism or sea-level changes. The best marker of inversion tectonics is represented by material shed from uplifting structures. Marginal troughs close to the southern margin of the Central European Basin contain sands, mostly derived from older Triassic to Lower Cretaceous clastic deposits. Inversion-related sandy to conglomeratic deposits allow provenance studies on the basis of clast and grain composition, heavy mineral analysis and zircon ages. The unroofing sequence was reconstructed for the Subhercynian Cretaceous Basin (von Eynatten et al., 2006) and the Bohemian-Saxonian Cretaceous Basin (Voigt, 2009; Hofmann et al., 2018; Nádaskay et al., 2019), with the main result that adjacent basement uplifts had been covered by Upper Paleozoic to Mesozoic sedimentary sequences.

Late Cretaceous marginal troughs of the North-Sea, accompanying the inverted Sole Pit Basin, Broad Fourteens Basin and the Central-Netherlands Basin, the Oldenburg and Münsterland Basins in northern Germany, and the marginal troughs at the Mid-Polish Swell and the Danish Basin were filled with authochthonous and re-deposited fine-grained deposits, marls, hemipelagic limestones, and chalks. They mostly preserve no particular provenance signal of the eroded succession, except reworked fossils (e.g. Wulff and Mutterlose (2019): Cenomanian calcareous nannofossils in Turonian limestones). The provenance signal of uplifted basement structures is also commonly obscured, because Permian to Mesozoic sediments covered them. The composition shows often only the signal of the basement which acted as the primary source of the eroded sediments (Niebuhr et al. 2014; Hofmann et al 2014, 2018; Nadaskay et al. 2019). Both in the Münsterland Basin and in the Subhercynian Basin, the main coarse clastic input during the early stages of inversion was apparently delivered laterally from other uplifted structures, and not from the main evolving highs related to the evolution of the marginal trough (Arnold, 1964; Voigt et al. 2006, von Eynatten et al. 2008). Coniacian sands in the Subhercynian Basin were probably redeposited from Lower Cretaceous sandstones covering the uplifting Calvörde High (part of "F" in Fig. 1) and its southeastern prolongation (Voigt et al., 2006; von Eynatten et al., 2008), whilst the basin margin in front of the uplifted Harz Mountains shows only a very thick marlstone succession in this period. This facies probably results from the removal of thick Upper Triassic to Jurassic claystones, which covered the Harz Mountains. The same is observed in the Münsterland Basin where Santonian sands were shed from the inverting Central Netherland Basin, while the inverting Lower Saxony Basin with its thick fine-grained Jurassic to Lower Cretaceous succession delivered the thick marly succession to the axis of the marginal trough (Arnold 1964). Although the sedimentary record allows a precise reconstruction of uplift rates and exhumation of uplifting structures in a few cases, recognition of early inversion in the sedimentary record is often ambiguous.

## 3.4 Slumps, slides and debris flow deposits

E. Voigt (1962, 1977) deduced a significantly earlier onset of deformation than previously inferred from unconformities and sediment composition by the observation of slumped and brecciated Turonian marly deposits close to the faulted margin of the Münsterland Basin (Osning Thrust). The oldest affected deposits are of Middle Turonian age and the slumps were initiated during the late Turonian or early Coniacian. Similar slumps and sedimentation anomalies occur frequently in the chalk of western Europe. They were described from the North Sea Basin, the Danish Basin and the Anglo-Paris Basin (Hardman, 1982; Bromley and Ekdale, 1987; van Buchem et al., 2017; Lykke-Andersen and Surlyk, 2004, 2007; Arfai et al 2016). Resedimentation is particularly common in Coniacian and Campanian deposits (Kennedy, 1987; Mortimore and Pomerol, 1997; Mortimore et al., 1998; Mortimore, 2011). The oldest occurrences of slumps and slides were reported in deposits of late Turonian age (Bromley and Ekdale, 1987; Arfai et al., 2016). Outcrops in the Weald Anticline (Sussex, Dorset) of the Anglo-Paris Basin show additionally indications of tectonically induced resedimentation in the chalk (Mortimore and Pomerol, 1991) starting in the Middle Cenomanian.

Submarine slumps develop on slopes of about 3-4° inclination in marly sediments if shear strength is exceeded by gravitational forces (e.g., Embley, 1982; Hance, 2003). Especially, unconsolidated, water-saturated mud is prone to such deformation processes. If additional loading of sediments results in pore water overpressure, or cohesion is low, a few degrees of steepening are sufficient to trigger mass flows. As the origin of mass flows depend on the shear strength of unconsolidated deposits, their initiation requires higher angles in pure chalk and hemipelagic limestones than in cohesive clay-rich sediments, but again the steepening has to be above 3° (Hance, 2003). Unconsolidated sandy deposits form sediment avalanches, resulting in turbidites, if the angle of repose is exceeded. Mass flows are therefore especially abundant in marly and clay-rich hemipelagic deposits (e.g., Hance, 2003). Slumps and debris flows at the active northern margin of the Münsterland Basin involve partly cemented hemipelagic limestones, evidenced by isolated angular clasts of varying size in marly breccias, prove that the inclination of the basin floor was probably in the order of several degrees. They therefore post-date the onset of inversion.

## 3.5 Changes in sediment thickness as evidence for basin formation

Flexure and subsidence of an elastic crust under a tectonic load immediately creates new accommodation space (Nielsen and Hansen, 2000; Hindle and Kley, this issue). If this space is completely filled by deposits, enhanced sediment thickness directly reflects the onset of loading and thus basin inversion. If sedimentation rates are low and the basin deepens without compensation of sediment accumulation, only subdued facies and thickness changes may show the onset of basin inversion. In the case of mild inversion, syntectonic deposition may persist on the tops of uplifting structures that may be revealed by a reduced thickness in comparison to the neighbouring marginal troughs.

Several studies show that thickness variations of clastic deposits in inversion-related basins of Central Europe become evident before the onset of inversion determined from thermochronologic ages and provenance studies. This is the case for the Subhercynian Basin (T. Voigt et al., 2006; von Eynatten et al., 2008), the Münsterland Basin (Arnold, 1964), and the basin flanking the inverted Mid-Polish Trough (Krzywiec, 2006; Krzywiec and Stachowska, 2016). In comparison to other features taken as markers for the timing of basin inversion, the differentiation of sediment thickness is probably best suited to pinpoint the onset and end of inversion in central Europe. Therefore, we use in the following the sediment thickness variation of the marginal troughs to determine the onset of deformation during Cretaceous basin inversion in Central Europe more precisely. We will use several case studies and data from the Bohemian-Saxonian Cretaceous Basin, the Subhercynian Cretaceous Basin, the Altmark Basin and the basins bordering the inverted Lower Cretaceous Lower Saxony Basin: The Münsterland Basin and the hitherto unnamed Upper Cretaceous basin north of the Rheder Moor-Oythe Thrust Belt in the subsurface of Lower Saxony, which we designate as South Oldenburg Basin. Additionally, we will address the question whether all prominent basement anticlines developed concurrently or in a particular pattern.

## 4 Dawn of inversion based on thickness differentiation

### 4.1 Münsterland Basin

The Münsterland Basin represents the southern marginal trough of the inverted Lower Saxony Basin (Fig. 2). The monotonous Albian to Campanian basin fill reaches its highest thickness (>2000 m) close to the Osning Thrust (Arnold, 1964). Thickness of Coniacian to Campanian strata (the marly "Emscher Facies") increases towards the thrust significantly. Slides and slumps indicate Turonian to Coniacian uplift and synsedimentary deformation close to the thrust (E. Voigt, 1962; 1977). The thickness of the syn-inversion deposits increases towards the thrust although most sediments derived from the inverting Roer Valley Graben and Central Netherland Basin in the west (Gras and Geluk, 1999), and from the southern margin of the Cretaceous sea (Fig. 1). As the northern margin of the Münsterland Basin was tilted and even partly overturned by the displacement along the Osning Thrust, the increasing thickness towards the central segment of the Osning Thrust can be demonstrated even in surface outcrops (Lehmann, 1999, Wilmsen et al. 2005, S. Voigt et al., 2008). Thickness differentiation occurred already in the Cenomanian and Turonian with the same depocenters as in the Coniacian to Campanian (Arnold, 1964). Sedimentation rates, however, are much lower than in the Coniacian and Santonian (Lehmann, 1999; S. Voigt et al., 2008), approximately 20 m/My. As the contour of the Cenomanian basin is identical to the structure of the inversion-related Coniacian to Campanian basin, a Cenomanian start of inversion is only indicated by increasing sediment accumulation, although no evidence for redeposition from the rising swell of the inverted Lower Saxony Basin has been observed.

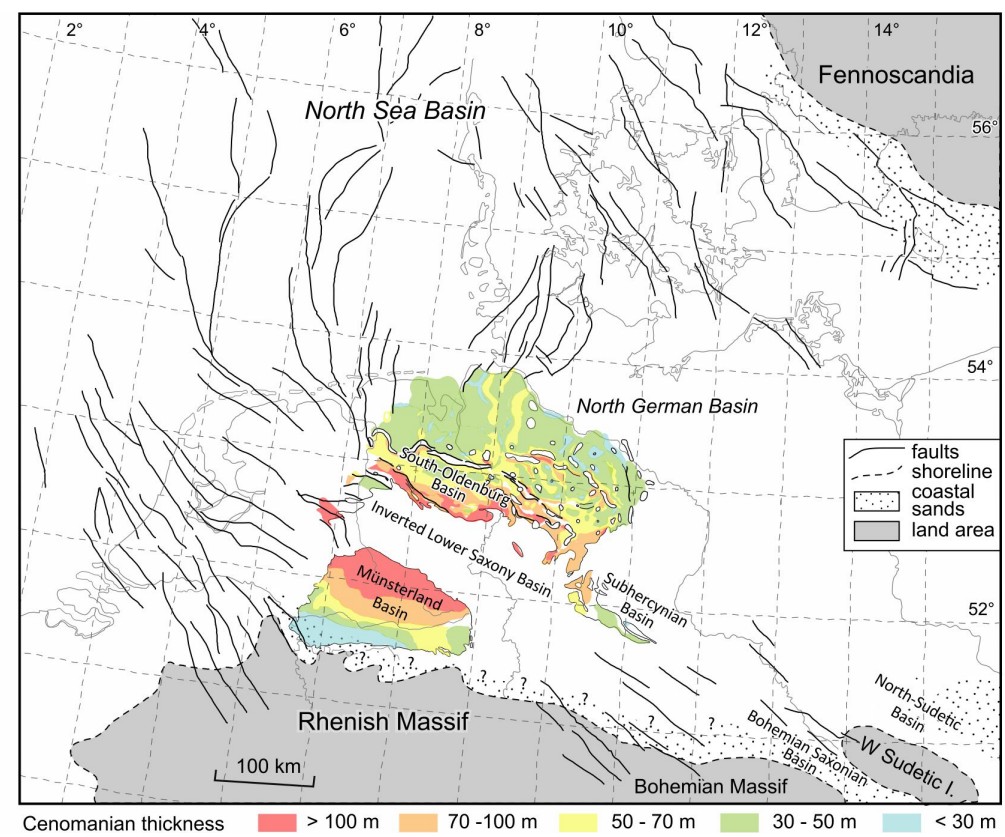

**Fig. 2: The southern and the northern margin of the inverted Lower Saxony Basin show enhanced thickness of Cenomanian deposits, indicating higher subsidence. In these fault-bounded symmetric marginal troughs, thickness of the complete late**
**Cretaceous succession exceeds 2000 m. Enhanced thickness occurs also in the peripheral sinks of salt diapirs in the North German Basin north of the South-Oldenburg Basin (modified from S. Voigt et al., 2008; compiled from Baldschuhn et al., 2001; Arnold et al., 1964 and Frieg et al., 1990)**

### 4.2 South-Oldenburg Basin

The South-Oldenburg Basin is part of the North German basin (Fig. 2) and evolved as a depocentre during the Late Cretaceous north of the inverting Lower Saxony Basin. The Jurassic to lower Cretaceous basin fill of the Lower Saxony Basin was uplifted several kilometres during the Late Cretaceous (Senglaub et al., 2005). The northern margin of the inverted Jurassic to Lower Cretaceous Lower Saxony Basin is marked by a system of thrust faults forming the Rheder Moor-Oythe thrust system (Fig. 3). These thrusts developed from the reverse reactivation of a swath of normal faults

accompanying the northern margin of the Lower Saxony Basin, a large Jurassic to Lower Cretaceous graben system (Baldschuhn et al., 1991; Kockel, 2003). The northern foreland (South-Oldenburg Block; Pompeckj Block) is characterised by a strong influence of salt diapirs on deposition, starting no later than the Jurassic and probably already during the Triassic

(Kockel et al., 2003; Warsitzka et al., 2019). Rising salt domes and subsiding peripheral sinks around those diapirs also influenced the general pattern of Late Cretaceous thicknesses. The facies pattern of northern Germany is dominated by chalk in the north and hemipelagic coccolithic to calcispheric limestones in the south, well-investigated with respect to typical log-patterns and biostratigraphy in boreholes (Baldschuhn and Jaritz, 1977; Koch, 1977, Wilmsen 2003).


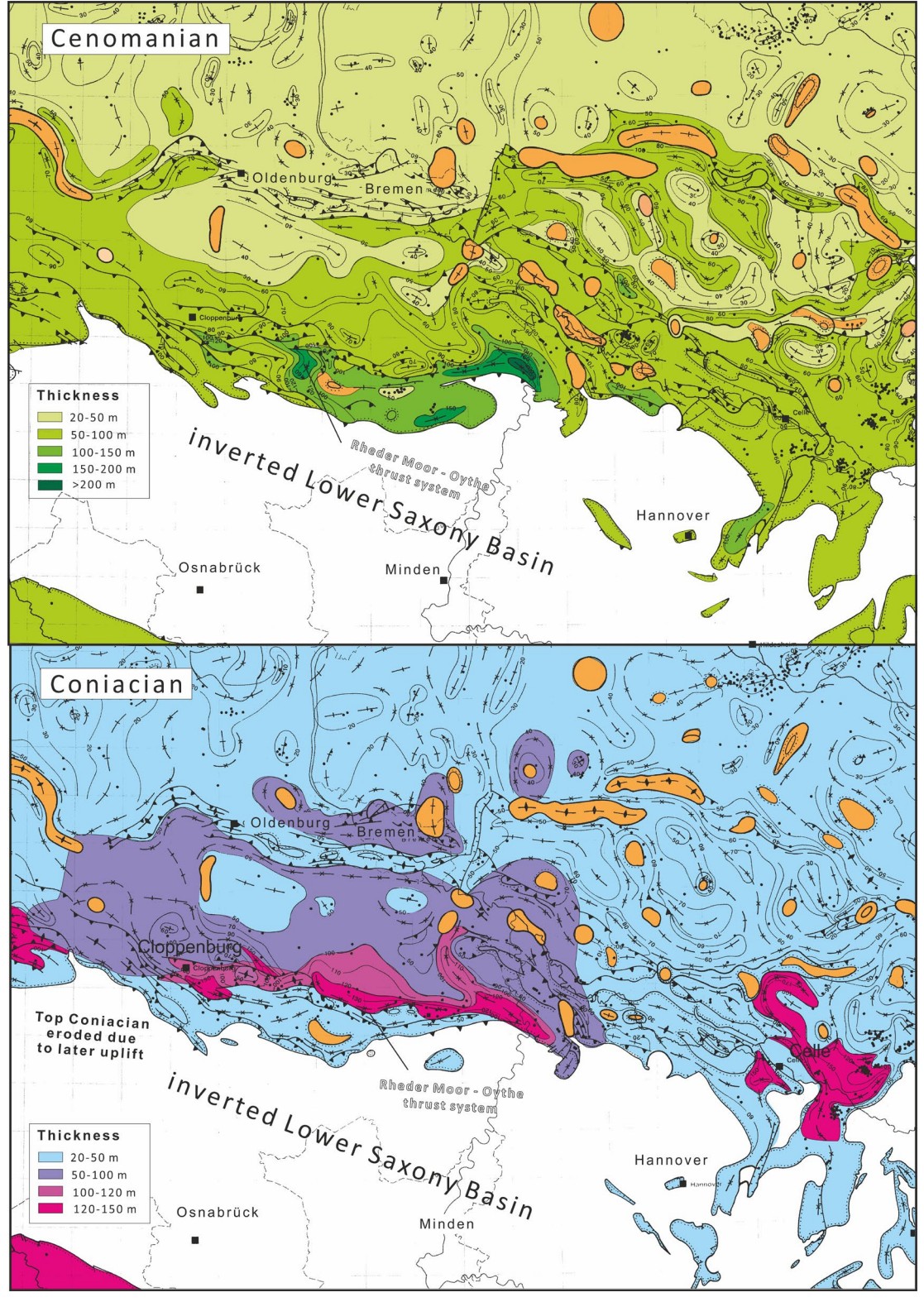

**Cenomanian**

Thickness
- 20-50 m
- 50-100 m
- 100-150 m
- 150-200 m
- >200 m

Oldenburg
Bremen
Cloppenburg
Celle

*Rheder Moor - Oythe thrust system*

*inverted Lower Saxony Basin*

Osnabrück
Minden
Hannover

**Coniacian**

Top Coniacian
eroded due
to later uplift

Thickness
- 20-50 m
- 50-100 m
- 100-120 m
- 120-150 m

Oldenburg
Bremen
Cloppenburg
Celle

*Rheder Moor - Oythe thrust system*

*inverted Lower Saxony Basin*

Osnabrück
Minden
Hannover

**Fig. 3: Detailed thickness maps of the northern margin of the inverted Lower Saxony Basin show that the syn-inversion Coniacian thickness distribution of the marginal trough already developed during the Cenomanian (modified from a thickness map of Baldschuhn et al., 2001). The shift of the basin axis of the marginal trough to the north can be explained by the propagation of thrusting towards the basin. Additionally, salt migration in the surroundings of salt diapirs (orange colour) created local highs and related local depocentres.**


Hemipelagic to pelagic deposits characterize the facies of the paired marginal troughs on both sides of the inverted Lower Saxony Basin. Chalk and hemipelagic limestones with upward-increasing marl content prevail, while coarser-grained deposits are absent. Most of the marls probably derived from re-deposited Jurassic and Lower Cretaceous sediments, because the Lower Saxony Basin fill was primarily composed of limestones, marl- and claystones. The thickness of Coniacian and

Santonian deposits in the adjacent marginal trough (South-Oldenburg Basin) increases towards the inverted normal faults of the graben structure of the Lower Saxony Basin. In the marginal trough, it attains about three times the background sedimentation thickness (Fig. 3). A key observation is that Turonian and Cenomanian deposits reflect already the same basin centres as the Coniacian to Campanian succession although thicknesses remain low (Fig. 3). The complete thickness of Cenomanian deposits varies between 50 to 200 m in the marginal trough, with a clear tendency to higher thicknesses in front

of the thrust system, while the Cenomanian thickness on the stable foreland remains between 20 and 50 m. The slightly varying thickness in the foreland is caused by salt migration. Comparison of thickness maxima shows that the zone of maximum thickness migrates trough time away from the inversion structure (fig. 3). The Coniacian thickness atop the southernmost thrust sheet in the South Oldenburg Basin is reduced. This observation probably indicates a successive activation of thrusts, propagating into the basin (thin-skinned tectonics), but needs further investigation.


### 4.3 Subhercynian Basin

The Subhercynian Basin contains a more than 2000-2500 m thick succession of Late Cretaceous sediments, which form a symmetric trough in front of the overthrust northern margin of the Harz basement anticline (T. Voigt et al., 2006; T. Voigt et al. 2009). Thickness of deposits is highest close to the thrust front. Sedimentation starts above a regional unconformity, which formed during the global Cenomanian sea-level rise. High sedimentation rates occur during the Coniacian to

Santonian (T. Voigt et al., 2006), but the first enhancement of thickness in the marginal trough in front of the Harz mountains is observed already in the Middle Turonian (Karpe, 1973, T. Voigt et al., 2006).

Cenomanian to Lower Coniancian thickness data of the Subhercynian Basin (fig. 4) were obtained from borehole logs (SP and GR) and corrected to dip. All sections show the general log pattern of northern Germany (Baldschuhn and Jaritz, 1977).

Therefore, a good correlation of sedimentary units is possible and stratigraphic gaps are well apparent, partly supported by sedimentary features and inoceramids in the cores (Karpe 1973). No boreholes reached the base of the Cenomanian in the

central marginal trough; therefore, the isopach map only displays a decreasing thickness trend to the southeast, not influenced by the Harz Mountains (Fig. 4).

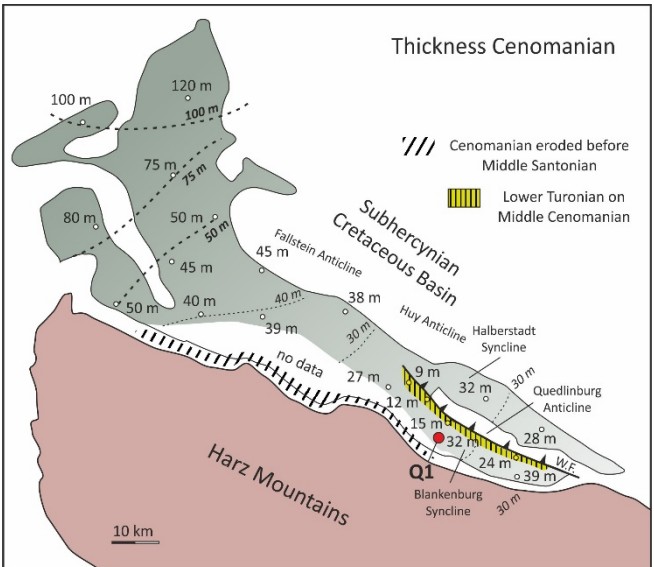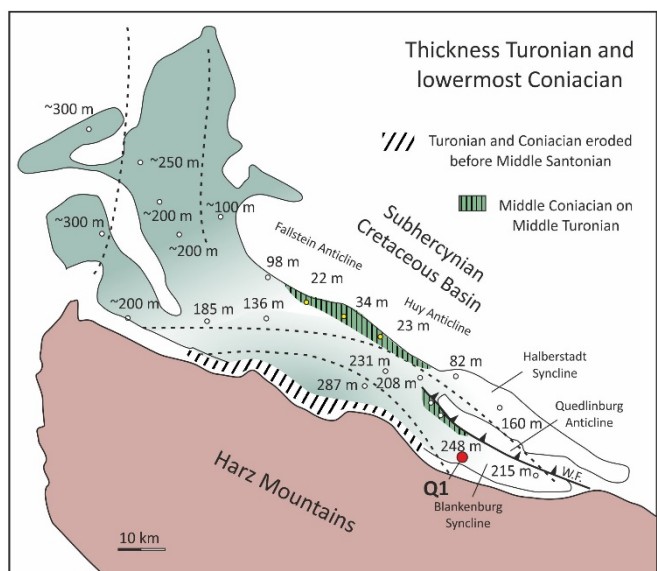


**Fig. 4: The first evidence of tectonic activity in the Subhercynian Cretaceous Basin is provided by strongly reduced thickness of Cenomanian deposits along the southern margin of the Quedlinburg anticline, the Westerhausen thrust (W.F.), which represents the master fault of an inverted early Cretaceous half-graben. The Turonian isopach map reflects the formation of a symmetric marginal trough in front of the Harz uplift and the continued uplift of the northern basin margin, accompanied by reduction of**
**Turonian thickness and erosion of a major part of the succession prior to the Mid-Coniacian transgression.**

The borehole Quedlinburg 1 (Q1), which is situated close to the thrust front but at the southeastern edge of the marginal trough, does not show an increased thickness of the Cenomanian (32 m) in comparison to the overall trend (Fig. 4).

The most striking evidence for a Cenomanian onset of compression in the Subhercynian Basin comes from an intra-basinal
structure. The Quedlinburg anticline represents a former half graben, which formed during the Early Cretaceous. The master fault of the graben about 40 km long became re-activated as a thrust/reverse fault during late Cretaceous inversion. Along the fault, at the margin of the adjacent syncline, Lower Turonian limestones cover middle Cenomanian marly deposits (Karpe, 1973). While the thickness of the lower and middle Cenomanian is similar to adjacent sections, the upper Cenomanian is missing or condensed. This points to a late Cenomanian activity of the thrust fault. During the Turonian and early Coniacian,
the structure remained active, but later erosion removed the evidence of tectonic activity close to the thrust. Nevertheless, at the western tip of the Quedlinburg anticline, the complete Turonian succession is preserved; hardgrounds and reduced thickness prove further activity of the thrust. Simultaneously, the Fallstein and Huy anticlines at the northern basin margin started to grow (Fig. 4, expressed in a significant unconformity of Middle Coniacian on Middle Turonian sediments and strongly reduced thickness in the Coniacian (Kölbel, 1944; T. Voigt et al., 2004).

## 4.4 Bohemian-Saxonian Cretaceous Basin

The Bohemian-Saxonian Basin is bordered by a significant post-depositional thrust (Lusatian Thrust) underlying the basement uplift of the Lusatian-Sudetic High. This thrust cuts through both coastal and hemipelagic deposits and clearly developed after the Coniacian (Fig. 5). Apatite fission track data, facies distribution and thickness data show, nevertheless, that the central segment of the fault already influenced sedimentation during Cenomanian and Turonian times (Seifert, 1955; T. Voigt, 2009; Lange et al., 2006; Danišík et al., 2010), probably by creating a fault-propagation fold (Voigt, 2009). Detritus derived from the exhumed Permian to Jurassic cover of the Neoproterozoic to early Paleozoic basement of the Lusatian High indicates the inversion of a Mesozoic graben structure (Voigt, 2009; Nádaskay, 2019). The preserved part of the basin fill ends in the Coniacian, with the exception of some deeply subsided remnants of Santonian sediments in the Ohře Graben, a segment of the European Cenozoic Rift System which was active in the Oligocene to Miocene. Fission track data point to a maximum uplift and exhumation between 85 and 75 Ma (Santonian to Campanian; Lange et al., 2006; Käßner et al., 2020), indicating that only parts of the basin fill are preserved. A subsequent regional uplift, which ended about 40 Ma ago, is shown by a regional unconformity at the base of Upper Eocene (?) and Lower Oligocene deposits of the Eger graben, which cuts across both the Lusatian-Sudetic uplift and its marginal trough (e.g. Standke, 2008; Migon and Danišík, 2012). These post-inversion deposits cover the basement, Permian red beds and Cretaceous deposits; with early Santonian deposits as youngest strata. In the east of the Bohemian-Saxonian Basin, main tectonic events occurred during the Paleogene and led to the uplift of intrabasinal highs by several kilometres (e.g. Danišík, 2012; Sobczyk, 2020).

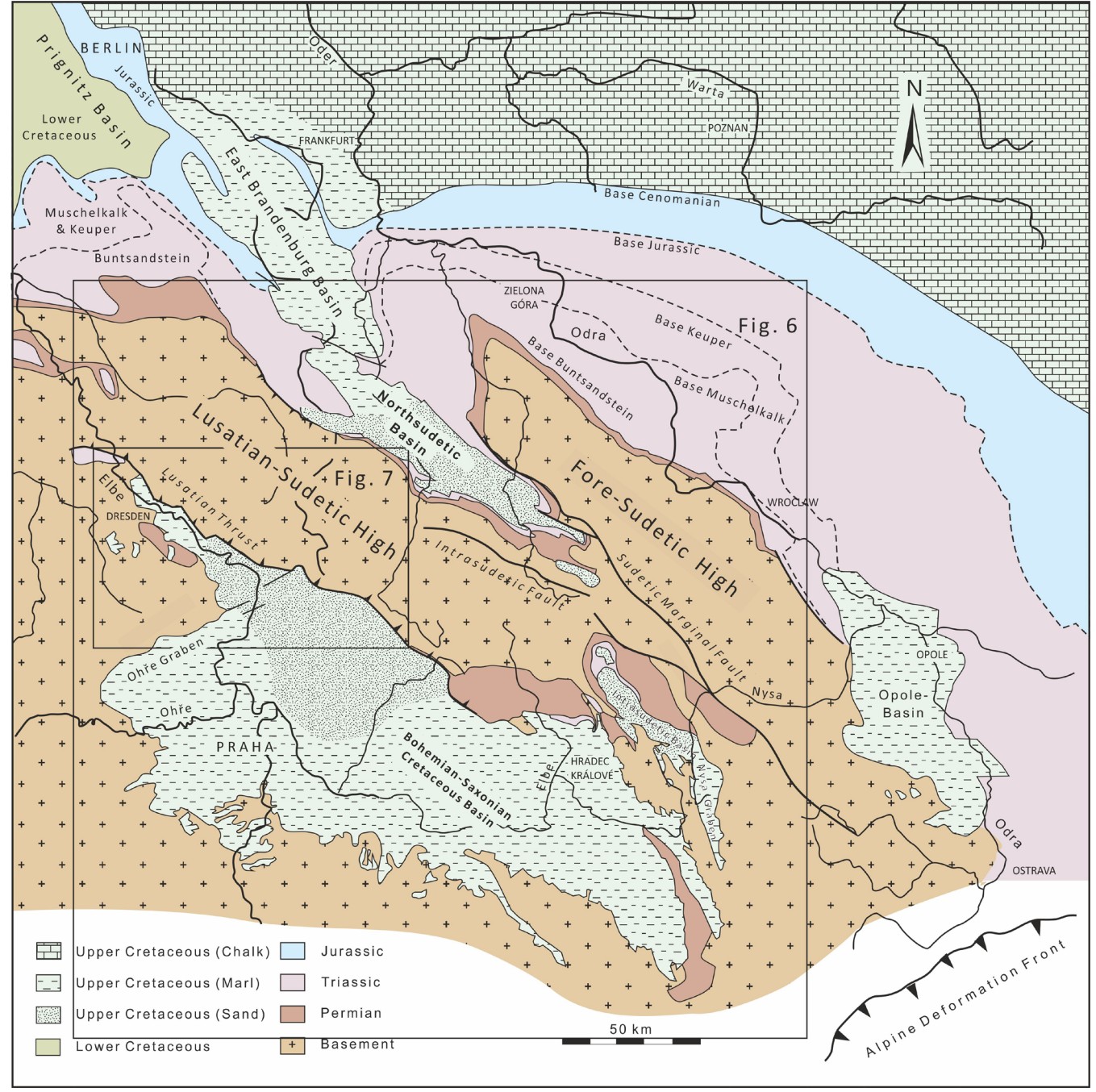

**Fig. 5: Late Cretaceous basins surrounding the Lusatian-Sudetic High show a strong confinement of clastic deposits to its margins. Regional Cenozoic uplift and denudation removed 1-4 km from both the Lusatian-Sudetic High and the related Cretaceous basins. Therefore, only remains of the primary basin fills were preserved, comprising deposits of Cenomanian to early Santonian age (modified from T. Voigt, 2009).**

Sedimentation within the marginal trough of the Bohemian Cretaceous Basin started in the Cenomanian, concurrently with the global sea-level rise. Deeply incised river valleys reflect a structured morphology with about 50 m of relief before the transgression (T. Voigt, 1998; Tonndorf, 2000; Uličný et al., 2009). The valley fills were preserved by the rising sea-level from the (Early?/Late Cenomanian to Lower Turonian, during a time-span of about 3 Ma. The pattern and evolution of these large paleodrainage systems were investigated by Uličný et al. (2009) in detail. Additionally, uranium exploration in the German part of the basin provided detailed data of the paleovalley pattern in the northwestern part. More than 1000 uranium exploration wells determined the paleovalley limits of the Niederschöna paleoriver, the Pirna paleoriver and the Hermsdorf paleoriver precisely (Tonndorf, 2000, Fig. 6). Considering the whole basin, a central water shed divided a northern paleodrainage system which was directed to the Boreal from a system draining towards the Tethys (Uličný et al., 2009; Fig. 6). The most striking feature of the valleys in the northern paleodrainage system is their orientation, because they reflect an inclination of the valley floors to the north. The Lusatian Thrust cuts at least four large and three minor paleoriver valleys discharging to the North. Uličný et al. (2009) assume a hypothetic principal stream running on the later exhumed Lusatian-Sudetic High parallel to the Lusatian Thrust collecting all the tributaries from the south. Nevertheless, there is no evidence for a river mouth in Lower Cenomanian deposits in the northern part of the basin, where Lower and Middle Cenomanian nearshore facies is preserved, so that a direct connection of the rivers to the Northsudetic basin can be also assumed (Fig. 6).

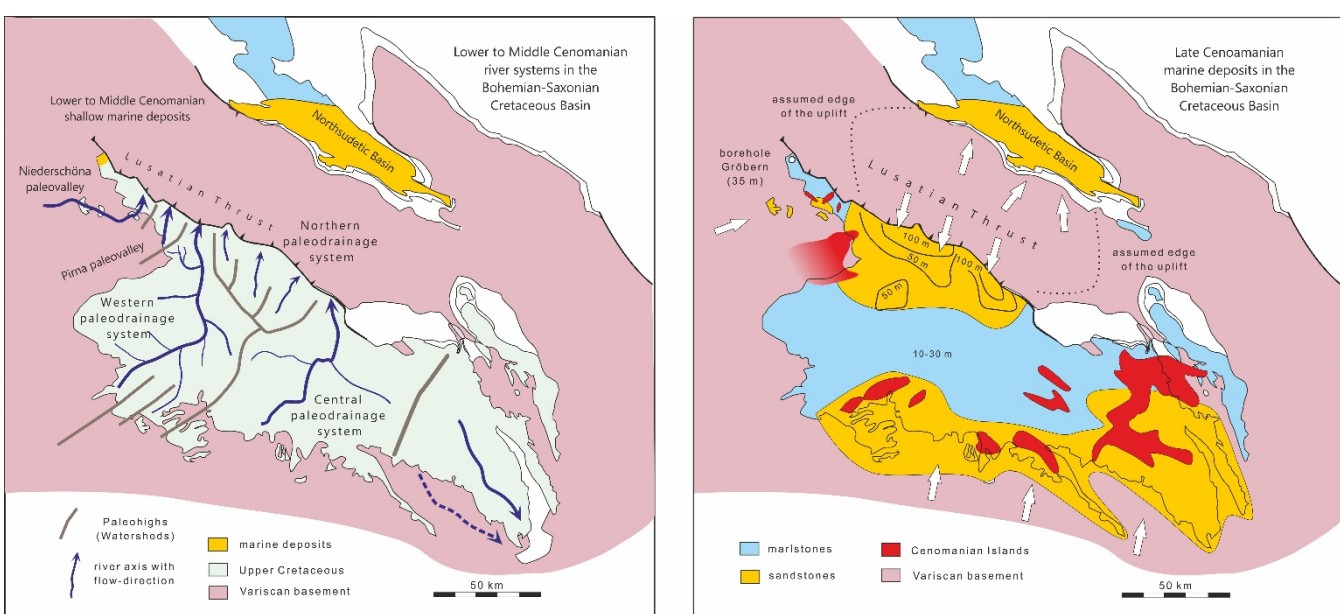

**Fig. 6: Deposition within the Bohemian Cretaceous Basin started in the early to middle Cenomanian with the filling of river valleys. Marine deposits were preserved at the northernmost edge of the basin. River orientation was directed to the north, towards the area which acted as source during late Cenomanian. The late Cenomanian basin configuration reflects the onset of uplift of the Lusatian-Sudetic High: the evolving marginal trough collects about 100 m of late Cenomanian sandstones compared**

 **to less than 30 m on the flooded shelf of the Bohemian platform. (paleodrainage pattern and Cenomanian facies and thickness after Uličný et al. 2009).**

The thickness of upper Cenomanian deposits still partly reflects the morphology of the pre-transgression landscape because the river valleys were gradually filled by clastic deposits eroded from the surrounding highs, while additionally an NW-SE
elongated depositional centre developed outside the ancient valleys. There, marine Upper Cenomanian deposits reach a thickness of up to 110 m (Fig. 7). This thickness increase is observed even on the former drainage divide between the paleovalleys of the central and northern paleodrainage system (Fig. 7). Cenomanian sedimentation rates are slightly lower (about 30 m/Ma) than those of the Turonian (50 m/Ma), but indicate a slow onset of basin subsidence. The hemipelagic facies on the northwestern edge of the basin shows also increased thickness compared to the Cenomanian of the Bohemian
platform outside the marginal trough. Upper Cenomanian deposits in the Gröbern borehole reach sedimentation rates in the order of 35-40 m/Ma (S. Voigt et al., 2006) compared to 5-15 m away from the basin axis on both the Bohemian platform and in the western Saxonian part of the basin. The higher thickness indicates probably the extension of the marginal trough further to the northwest, although facies belts remained stable.

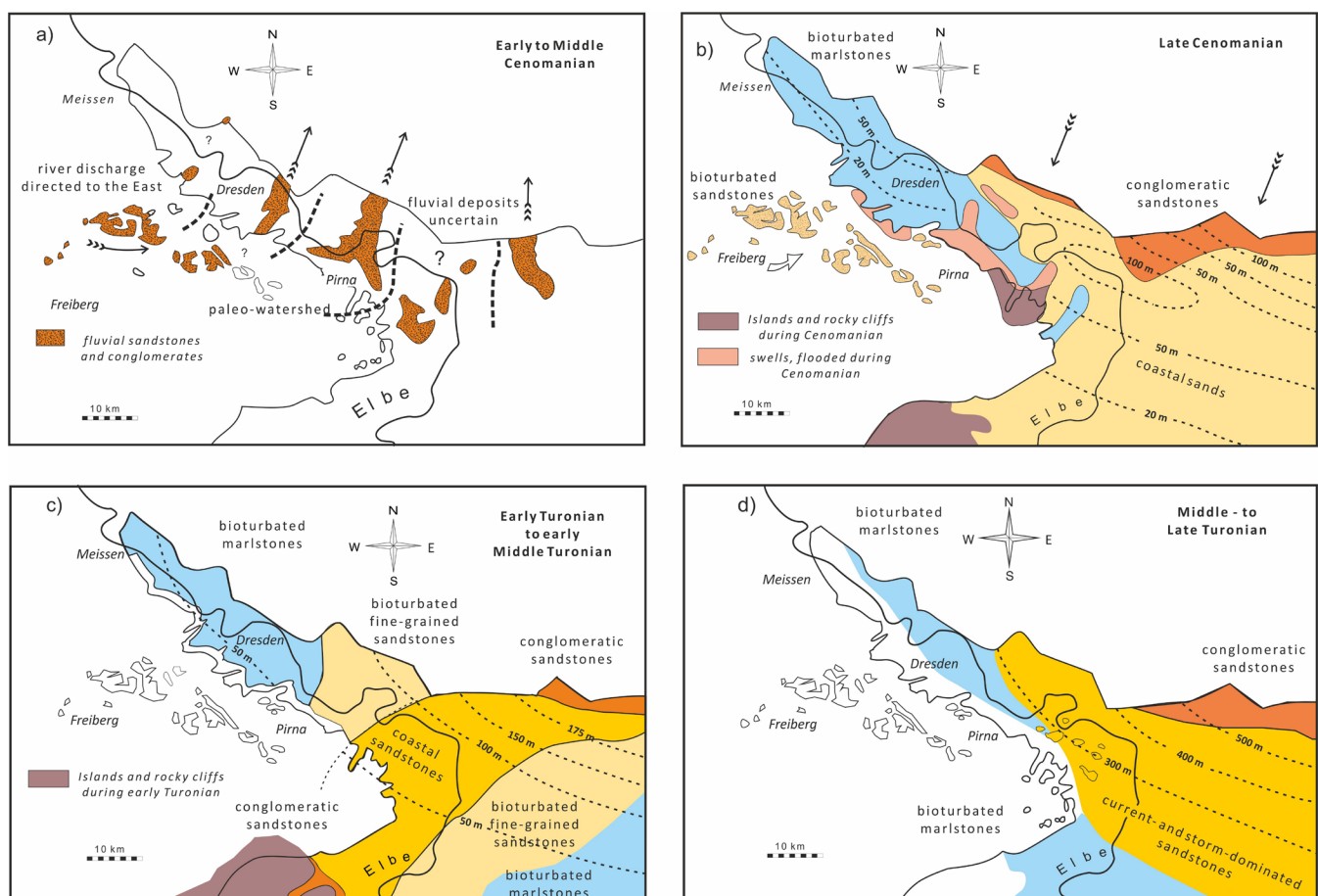


**Fig. 7: Detailed facies maps of the Saxonian part of the Bohemian Cretaceous Basin. Early to middle Cenomanian rivers discharge to the north. Distribution of sandstones in the Cenomanian and early Turonian reflects a marginal trough in front of the rising high, and thus the complete re-organisation of the basin configuration. Coastal sandstones of middle and late Turonian age mark the northwestern edge of the Lusatian-Sudetic High. The late Cretaceous to Paleogene Lusatian Thrust cuts through the**
**basin margin and distal deposits. The data of the isopach maps are derived from numerous boreholes and geological maps.**

The facies distribution clearly indicates a major source area in the northeast, reflected by sandstones and conglomerates close to the northeastern basin margin (Fig. 7) and thus, reversing the drainage direction during early and middle Cenomanian. This basin axis is nearly identical with the marginal through which started to develop slowly during the Turonian and lower
Coniacian (Fig. 7) but consisted of several subbasins (Uličný, 2001, Niebuhr et al 2019). A possible explanation is a separate evolution of several small uplifts, which later unified to form a single source area and the integration of the separated depocenters into one marginal trough. Alternatively, the oblique convergence phase observed by Navabpour et al. (2017) in small-scale structures, and which predates the frontal thrusting, could have induced the subsidence of oblique en-échelon subbasins.

Together with the significant change of the basin floor morphology, this change during the late Cenomanian indicates a complete reorganisation not only of the depositional system but also of the stress field within the basin. Regardless of whether the hypothetic NW-directed trunk stream of Uličný et al. (2009) existed or not, the appearance of a large source area in a direction downstream of the former drainage indicates the uplift of a former topographic low.

**5        The dusk of Late Cretaceous basin inversion**

The end of basin inversion/basin uplift in Central Europe is even more difficult to define than its onset because the region was affected by large scale regional uplift which continued, possibly in different uplift phases until the Paleogene, and even longer south of the inverted Lower Saxony Basin and uplifted Harz Mountains (von Eynatten et al., this issue). Due to this event, some thermochronological data show a continuation of uplift up to 60 or even 50 Ma (von Eynatten et al., 2019). The

end of the tectonic activity of a single structure can be shown if the structural configuration of highs and lows changes; new depositional centres evolve or formerly active structures and folds are covered by younger sedimentary units. If no deposition occurred during reconfiguration, for example because the whole region was above base level or the area was affected by later uplift, the recognition of a new stress field and differentiation between regional uplift and inversion remains ambiguous. Only in the subsurface of the deeply subsided Central European Basin, a complete succession of syn- and post-

inversion deposits is preserved, such as in the inverted Danish Basin, in the North Sea and in the Dutch basins. At the border of the inverted Danish Basin, a rapid shift of the basin axis and therefore the end of basin inversion occurred in the Danian, followed by moderate further uplift caused by crustal relaxation (Nielsen et al., 2007). In the Roer Valley Graben and in other inverted basins in the Netherlands, Late Cretaceous inversion ended in the latest Maastrichtian, evident from the cover of uppermost Maastrichtian and Danian chalks on top of the inverted axis (Deckers and van der Voet, 2018). The Polish part

shows a differentiated evolution: deformation seems to have continued until the Paleocene on the northern side of the Mid-Polish Anticlinorium (Krzywiec, 2006), while its south side experienced regional uplift, expressed by a marked unconformity across the marginal trough and the swell below an Eocene succession.

A similar situation is observed in central Germany, where Eocene to Oligocene deposits cover large areas of the structures resulting from Late Cretaceous inversion, including most of the basement uplifts. Only a few places in northern Germany

allow the recognition of the basin configuration change. In general, the youngest deposits preserved within the marginal troughs are of early Campanian age (Subhercynian Basin, South-Oldenburg Basin, East-Brandenburg Basin). Deposition in the Münsterland Basin continued until the late Campanian. In all these basins, thermochronological data, erosional unconformities and composition of the basin fill prove a younger uplift which involved both the source area and the adjacent marginal trough. This long-lasting unconformity, covered by Eocene to Oligocene sediments, is partly still visible in the

recent morphology. In the Erzgebirge, in the Lausitz and in the Harz and their forelands, peneplains of Late Cretaceous to Paleogene age are still preserved (e.g. Standke, 2008; Blumenstengel and Krutzsch 2008).

The time gap to younger deposits above those unconformities spans mostly more than 30 Ma, due to the absence of Late Campanian to Paleocene deposits. This is partly caused by a significant sea-level fall, which occurred during this period (Haq, 2014), but is mainly generated by regional uplift of those structures ("Laramide uplift"). Maastrichtian and Paleocene deposits are therefore rarely preserved in Central Europe, while Maastrichtian and Danian deposits occur on top of several inverted basins and their flanks in the Netherlands (Roer Valley Graben, Broad Fourteens Basin, Central Netherlands Basin, Dutch Central Graben) according to van der Molen et al. 2005; Deckers and van der Voet 2018). The widespread deposition on top of formerly inverted structures reflects there the end of Late Cretaceous inversion. Remains of similar deposits occur only at the margins of active diapirs and in a few narrow basins which do not reveal the configuration of former, late Cretaceous marginal troughs. These remnants witness an extended facies belts of a shallow shelf from continental to shallow marine environments, which grade into the hemipelagic and pelagic chalk environments of the central basins (Diener, 1968; Voigt, 2008). Their patchy occurrence indicates a nearly flat surface across both inverted highs and marginal troughs.

To better constrain the timing of the formation of this significant unconformity, we consider the examples of the Altmark Basin and the inverted Lower Saxony Basin with the unconformably overlying Campanian deposits of the Damme Syncline, and the Prignitz High, which represents the less inverted prolongation of the Lusatian-Sudetic High (Fig. 1). The Subhercynian Basin and the Harz Mountains are taken as an example of an inversion structure with poorly constrained end of contraction and are therefore only briefly discussed.

## 5.1 Dusk of Cretaceous and dawn of Paleogene inversion in the Altmark Basin

The Altmark Basin is an elongate, about 60 km long and only 15 km wide marginal trough (Fig. 8), which formed north of the uplifted Calvörde Block above a salt detachment linked to the Gardelegen Fault (Schulze, 1964; Kossow, 2001; Malz et al., 2019). AFT ages from the Permian sandstones of the Flechtingen High, a part of the exhumed basement of the Calvörde High, suggest rapid cooling around 70 Ma (Fischer et al., 2012), confirming the overall pattern of Late Cretaceous syntectonic basin formation in Central Europe. The thermochronological age is, however, not in good agreement with the accompanying marginal trough north of the Gardelegen Thrust, which preserves a syncline filled by a more than 700 m thick succession of syn-inversion deposits very similar to those of the Subhercynian Basin, indicating main inversion between 85 and 75 Ma. A late anticline divides the basin into two parts. Increased subsidence in comparison to neighbouring basins began slowly in the Turonian (Cenomanian thickness has not been studied yet in detail) and reached its maximum during the Coniacian to early Campanian. The youngest preserved deposits are of early Campanian age in the central marginal trough and reach at least 450 m thickness (Schulze 1964). Close to the Gardelegen Fault, Santonian sediments contain conglomerates and sands derived from the exhumed Mesozoic cover of the Calvörde Block (Schulze, 1964). This indicates that the uplift of the Flechtingen High, which is the central part of the Calvörde Block and was thrust onto Mesozoic deposits along the Haldensleben reverse fault, postdates the exhumation of the greater structure which demonstrably acted as a source

area in the Santonian (85-82 Ma). To obtain a well-constrained exhumation age, the uplift of the Flechtingen High relative to
the Calvörde Block must be about of additional 2-4 km, because the PAZ of the preceding uplift is not preserved.

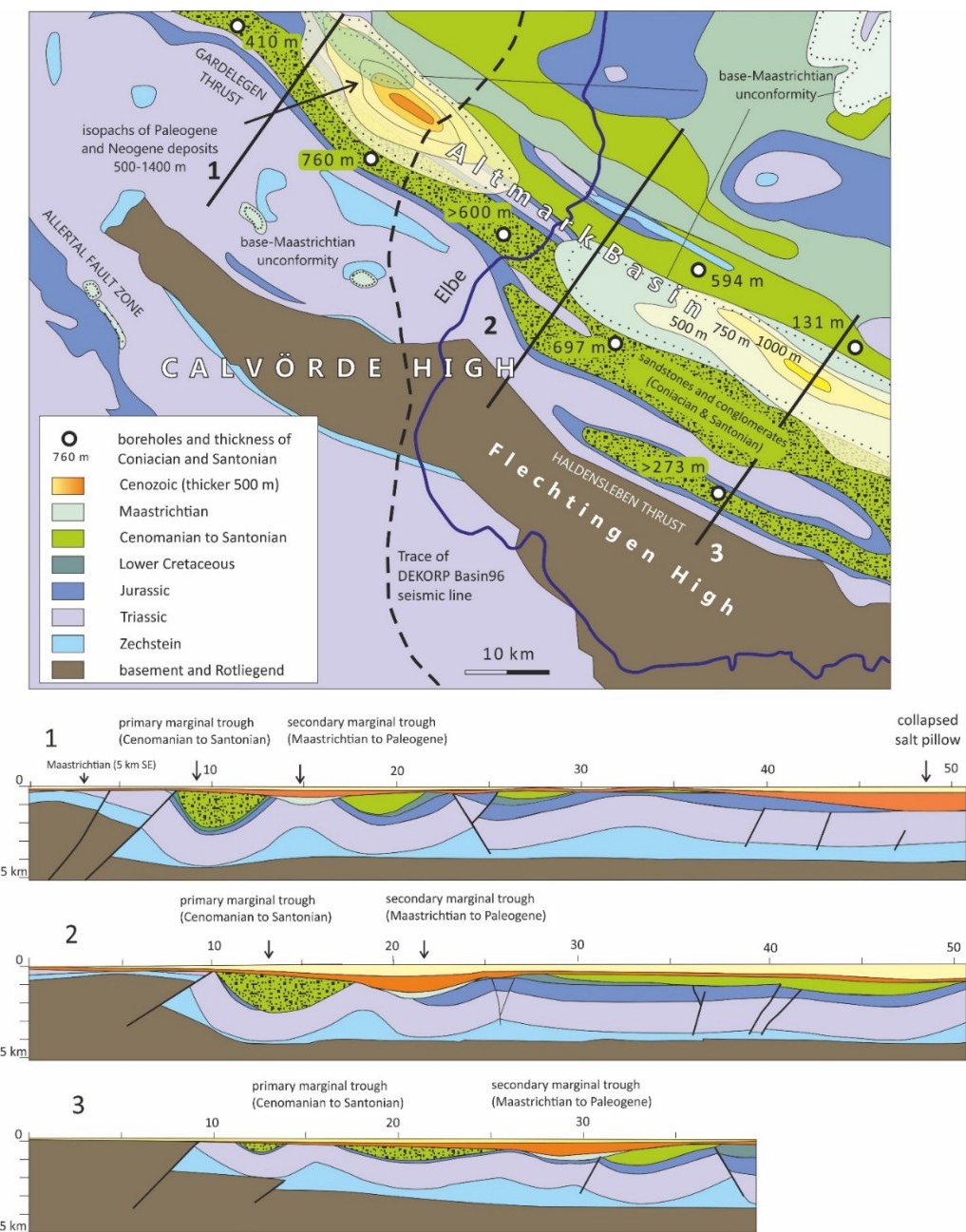

**Fig. 8: The Altmark Basin represents a narrow marginal trough north of the uplifted Flechtingen High. Deposition within the**
**basin, which is dissected by a salt-intruded anticline, ended in the early Campanian. A shallower basin developed north of the**

**Altmark Basin above an unconformity cutting across the highs and basins at the base of the Maastrichtian. Note that Paleogene deposits reflect the same depocentres as the Maastrichtian and are therefore considered as secondary marginal troughs. The map was constructed on the base of Malz et al. (2019), Schulze (1964) and the interpretation of borehole data. Location in Fig. 1.**

Zircons and volcanic quartz grains, resulting from the erosion of the Permian volcanic basement of the Flechtingen High appear late in Maastrichtian sands (Walbeck, Weferlingen), south of the uplifted structure at the Allertal Fault zone (Götze and Lewis, 1997). The provenance signal confirms the modelled AFT ages precisely (Fischer et al 2012). These Maastrichtian shallow-marine sands rest unconformably on Triassic deposits, again indicating the covering of an inverted structure, which was eroded and started to subside again. The total uplift of the region since 70 Ma (Maastrichtian) is less

than 2 km, indicating that the post-inversion configuration is more or less preserved. This inference is also supported by the nearly complete cover of the area by early Oligocene deposits (Blumenstengel and Krutzsch, 2008). Especially the base of the Rupelian transgression is a good representation of the base-level. Elevation changes of this marker horizon indicate post-Rupelian tectonic movements, salt flow or both.

The regional seismic section DEKORP Basin 96 (e.g., DEKORP Basin Group, 1996; Kossow, 2001) and boreholes drilled

for gas exploration allow to reconstruct the structural pattern. The succession of the marginal trough containing Cenomanian to Santonian deposits is bounded in the south by the Gardelegen Thrust Fault and in the north by a thin-skinned contractional salt anticline, which developed after deposition of the basin fill (Malz et al., 2019). Borehole stratigraphy and reflection patterns in the seismic section indicate a varying proportion of preserved strata (Schulze, 1964; Musstow, 1975). The shortened marginal trough was uplifted and eroded without further deformation. The flat erosion surface was tilted and can

be traced beyond the extent of Cretaceous deposits onto the Calvörde Block (Malz et al., 2019). It is inclined to the north and forms the flank of a new depocenter, which developed north of the Cretaceous depocentre and covers the partly eroded salt anticline. The sedimentary succession above this erosion surface shows a progressive onlap, starting with continental to shallow marine Maastrichtian sands (Oebisfelde member of the Nennhausen Formation; 200-330 m), followed by a Paleocene (uppermost Danian) succession (Wülpen Formation; maximum 200 m), indicating slow subsidence of the trough.

Numerous boreholes document a saucer-shaped, symmetric structure (Fig. 8) of the secondary marginal trough. In comparison to the Late Cretaceous one, it is wider and shallower than the primary marginal trough. Thanetian sandy deposits cover both the Calvörde High and the complete foreland with the marginal troughs (Blumenstengel and Krutzsch, 2008). The difference in structural elevation between the marine Maastrichtian on top of the Calvörde High and in the syncline is 500 to 1000 m, indicating Paleogene subsidence of the Altmark Basin. After late Paleocene erosion, upper Eocene marine deposits

transgressed locally even onto the Flechtingen High, demonstrating the transition from uplift to subsidence there (Fig. 8). Nevertheless, the area was completely covered by marine deposits not before the Rupelian.

This structural situation matches the evolution of primary and secondary marginal troughs described by Nielsen et al. (2002) and Nielsen et al. (2007) from the inverted Danish Basin. However, the trough is deeper and situated closer to the inverted

structure. The described sudden shift of the basin axis occurred before the Maastrichtian and is thus like the inversion history of the Vlieland Basin (Deckers and van der Voet, 2018).

A differing interpretation of the secondary marginal trough could be that collapse of the salt-cored anticline with extrusion and marine dissolution of salt caused the newly created depocenter. However, the extent, the smoothness and the undisturbed succession above the suggested dissolution surface disagree with this interpretation, because sediment deposition would cease further dissolution by sealing. Regardless of this interpretation, the base-Maastrichtian unconformity is a prominent feature at many structures in the North German Basin such as at the western Allertal Fault zone (Lohr et al., 2007) and the Prignitz High (T. Voigt, 2015).

## 5.2    Dusk of Cretaceous inversion in the Subhercynian Basin

The preserved sediment column of the Subhercynian Basin ends in the Lower Campanian although fission track data suggest continuous erosion of the Harz Mountains during the entire Campanian and even into the Paleogene (von Eynatten et al., 2019). Those younger deposits were eroded before the Eocene, because deposits of this age are preserved close to the front of the Harz and at the borders of some anticlines at the northern margin of the marginal trough. Because the central Harz was covered by deposits of Oligocene age (König et al., 2011), inversion had apparently ended in the Eocene and only mild regional uplift affected the region subsequently (König et al. 2011; von Eynatten et al., 2019; Paul, 2019).

Late uplift involved both the basement uplift and the surrounding basins. The time gap between the last preserved Lower Campanian inversion-related deposits (~82 Ma) and the Eocene/Oligocene deposits (~34 Ma) within the Subhercynian Basin is approximately 40 Ma. A more precise time estimate of the basin configuration change is therefore not possible. However, both the Harz Mountains and its foreland show a significant peneplanation cutting across all lithologies of the uplifted block and the basin (König et al., 2011), which formed between early Campanian and Oligocene times. In the Harz Mountains, remains of Oligocene (Rupelian) deposits are preserved in karst caves within Devonian limestones of the Elbingerode complex (Blumenstengel and Krutzsch, 2008, König et al., 2011). They are about 140 m above the level of the Oligocene transgressive surface in comparison to the same stratigraphic horizon south and east of the Harz Mountains and indicate moderate uplift which was not accompanied by major erosion since then. While König et al. (2011) interpreted this elevation difference as an effect of renewed motion on the Harznordrand Thrust, Paul (2019) argued that the observed offset was the result of foreland subsidence due to salt dissolution at depth.

## 5.3    The Damme Syncline: The end of inversion in the Lower Saxony Basin?

The Damme Syncline is an erosional remnant of uppermost lower Campanian to Maastrichtian sediments of about 300 m thickness resting on the inverted Lower Saxony Basin (Fig. 9). Inversion of the Lower Saxony Basin was asymmetric, leading to the uplift of Triassic deposits and some small basement uplifts (Ibbenbüren High, Piesberg, Hüggel) to the surface

in the south. In the north, a lower degree of uplift is observed, resulting in the preservation of parts of the Jurassic to Lower

Cretaceous basin fill (Baldschuhn et al., 1991; Senglaub et al., 2005).

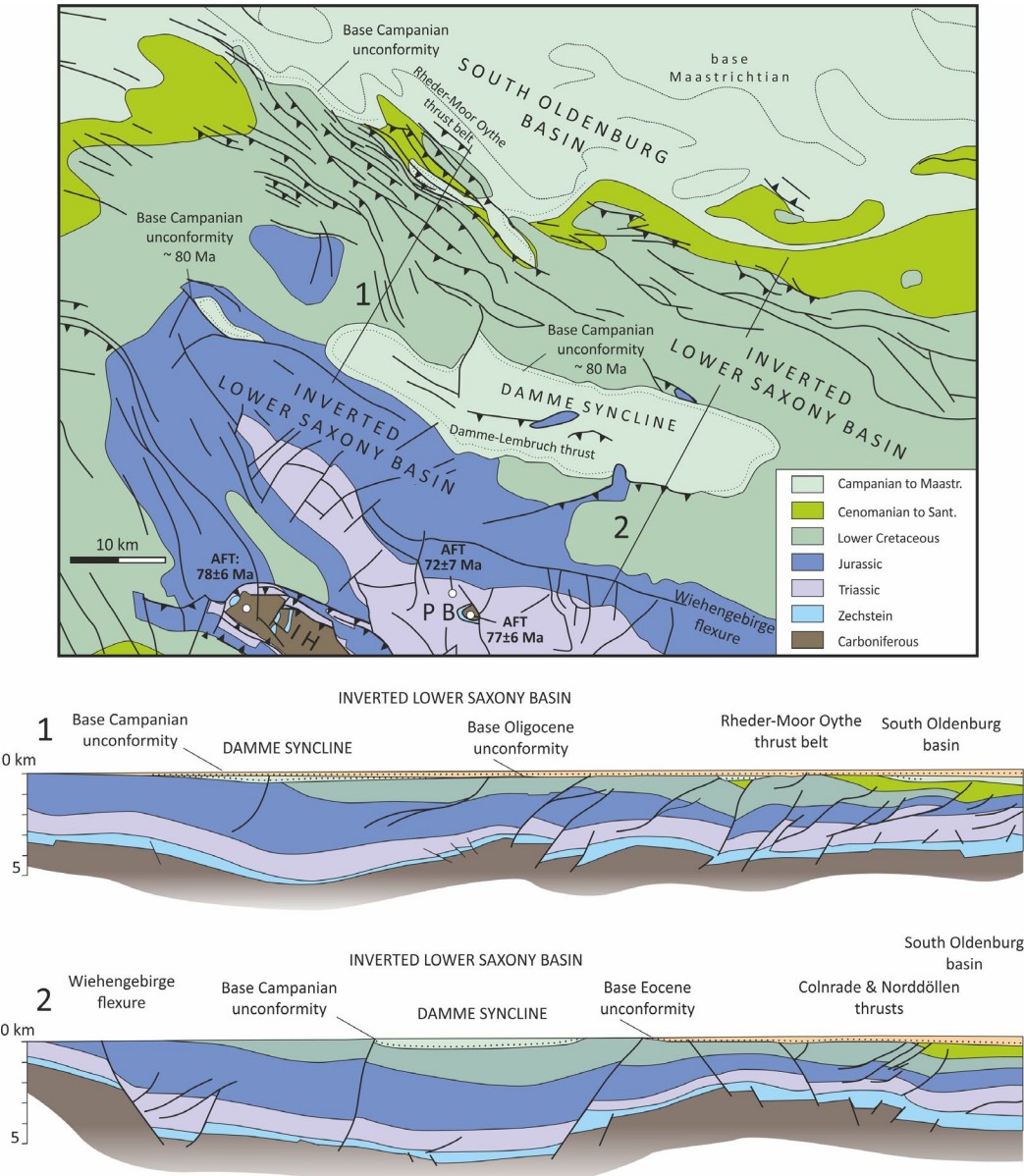

**Fig. 9: The oldest deposits on top of the inverted Lower Saxony Basin middle to upper Campanian bioclastic limestones resting transgressively on a peneplain cutting Jurassic and Lower Cretaceous deposits. They predate the uplift of the strongly inverted southern part of the Lower Saxony Basin and suggest that compression of the lithosphere ceased slowly. Deformation of the Damme Syncline and thrusting within Campanian deposits in the north suggest progressive deformation. IH, PB: Ibbenbüren High and Piesberg basement uplifts. Regional uplift prevented deposition of Paleogene sediments older than Oligocene both on the**

 **High and the adjacent Münsterland and South-Oldenburg basins. Map and cross-sections are based on Baldschuhn et al. (2001), AFT-data are from Senglaub et al. (2005).**

The syncline is gently folded and affected by a thrust (Damme-Lembruch Thrust) of about 200 m displacement, indicating post-depositional contraction (Fig. 9). The marine Campanian sediments unconformably cover deformed Jurassic and Lower
Cretaceous strata. Their deposition postdate the subsidence of the marginal troughs, which are flanked on both sides of the inverted basin and contain syntectonic basin fills of Cenomanian to early Campanian age. The Pompeckj Block on the north side preserved deposits of that age but with a chalk facies differing from the clastic succession on top of the inverted basin. The transgressive succession of the Damme syncline consists of bioclastic nearshore limestones and reworked ironstones at the base, followed by sandy marls (e.g., Mortimore et al. 1998). In contrast, the nearest upper Campanian units of the South-
Oldenburg Basin exhibit typical mid-to-outer shelf marine chalk facies, assumed to have been deposited in water depths between 100 and 150 m (e.g., Boussaha et al., 2017; Machalski and Malchyk, 2019). AFT cooling ages range between 72 ±7 and 78 ±6 Ma in the hanging wall of the adjacent Wiehengebirge flexure zone (Senglaub et al., 2005), in general covering the same time span as the sediments above the unconformity (Fig. 9). The southern Lower Saxony Basin acted as a source for the siliciclastic share of sediments in the Damme Syncline.
Eocene to Oligocene deposits cover both the inverted Lower Saxony Basin and the South Oldenburg Basin above a second unconformity and show that no major uplift affected this part of the inverted Lower Saxony Basin since the late Campanian. The weak folding of the first unconformity and the incipient Damme-Lembruch thrust demonstrate deposition in the same tectonic regime as during deformation of the underlying inverted Lower Saxony Basin. The preservation of these deformed late-inversion sediments indicates the absence of major erosion since the Late Cretaceous.


## 5.4 End of inversion tectonics at the Lusatian-Sudetic High

The inverted Lusatian-Sudetic Block is bounded by the marginal troughs of the Bohemian Cretaceous Basin and the Northsudetic Basin. Investigations of the Late Cretaceous to Paleogene basin evolution by different authors are mainly based on thermochronology and thermal maturity of the hanging wall and the footwall block (Käßner et al., 2020; Lange et al.,
2006; Danišík et al., 2010; Sobczyk et al.; 2020) and time constraints derived from geometrical relationships of strata, magmatic dykes and faults (Tietz et al., 2015; Coubal et al., 2014). The sedimentary record of basin inversion ends with remains of lower Santonian deposits, preserved in the central Ohře Graben, or with lower Campanian deposits, drilled in a syncline in the central North Sudetic Basin. A late, Paleogene, activity of a segment of the Lusatian Thrust was inferred by Käßner et al. (2020). AFT cooling ages (84-70 Ma) show an accelerated uplift during the latest Cretaceous. Younger ages
(ca. 40 Ma) in the area of the Krkonosze and the Jizera mountains provide evidence of Paleogene uplift. Renewed sedimentation started in the Oligocene with the formation of the Ohře Graben, which crosscuts the Lusatian Thrust and covers both the basin and the inverted Lusatian Massif (Coubal et al., 2015; Špičáková et al., 2000). A later, weak

reactivation of the Lusatian Thrust was reconstructed based on offset Oligocene tuffs (of 30-27 Ma age) by Tietz and Büchner (2015), although the orientation of the fault displacing the tuffs is oblique to the thrust. The end of inversion of the Lusatian massif is poorly constrained to a period of at least 40 Ma duration (Santonian/early Campanian to Oligocene), with the limitation that post-Cretaceous exhumation did not exceed 2 km (Käßner et al., 2020).

Thermochronology data derived from intrabasinal highs within the eastern part of the Bohemian-Saxonian Basin (Intrasudetic Basin) indicate partially fully reset AFT ages during late Cretaceous, thus proving that ranges of the Sudetes were initially part of the marginal trough (Danišik et al., 2012, Botor et al. 2019). The main rapid exhumation of these structures occurred in the Paleogene along N-S oriented thrusts, mainly between 63 and 40 Ma (Sobczyk et al., 2015; Sobczyk et al., 2020; Botor et al., 2019). The deep burial and formerly enormous thickness of the marginal trough of the Lusatian-Sudetic High is supported by recent unpublished maturity data of organic matter, derived from basal Cenomanian coals in Saxony, which point to a very thick filling of the marginal trough, with burial depths of >3-4 km during the Cenomanian–Campanian. The AFT cooling ages of deeply buried Upper Cretaceous sediments (46 Ma) are slightly younger than the highs and indicate regional uplift and erosion during Paleogene.

Better time constraints are found in the northwestern prolongation of the Lusatian-Sudetic High, the Prignitz High (Fig. 10). In this part, the Jurassic to Lower Cretaceous Basin shows only mild inversion of less than 1000 m. During Coniacian to Campanian uplift, the Prignitz High delivered clastic material into shallow marginal troughs north and south of the rising swell. As the uplift rates were low, marginal and even central parts were transgressed as is shown by the preservation of marine sediments in peripheral sinks of major diapirs on the swell (Haller, 1965; Musstow, 1976, Karpe 2008). To the southeast, the broad uplift of the Prignitz High is bounded by the deeply subsided Altmark Basin, which was predominantly filled with sands and marls from the narrow uplift of the Flechtingen High (Fig. 8). The facies distribution changed significantly from the Campanian to the Maastrichtian.

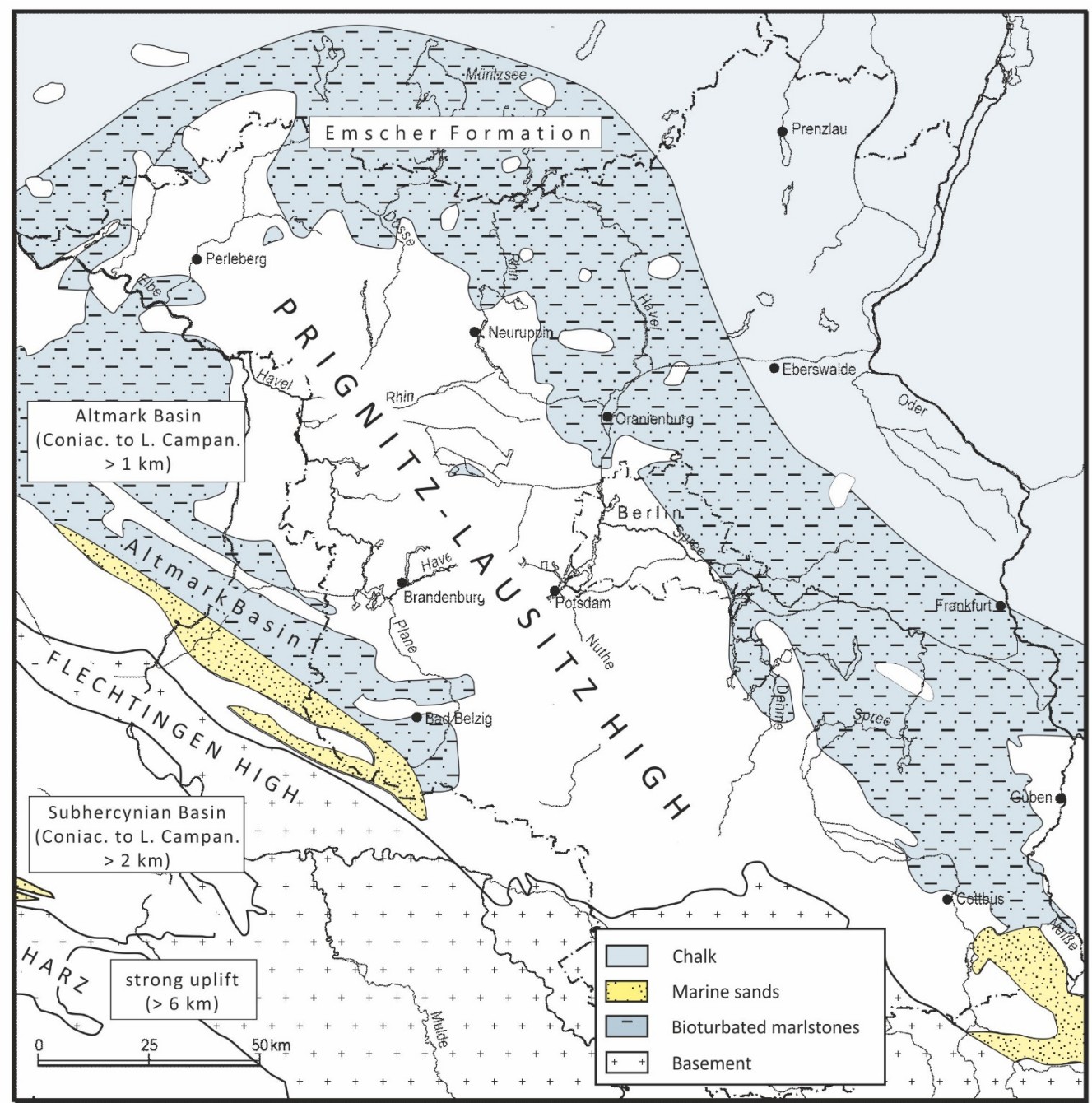


**Fig. 10: The Prignitz High represents a gentle late Cretaceous inversion structure in the north-western prolongation of the Lusatian-Sudetic High. Note the preservation of marly deposits in a peripheral sink of a diapir at the centre of the swell. During the main inversion phase, it was surrounded by a belt of marlstones derived from the erosion of Jurassic to Lower Cretaceous claystones. Marine sands were restricted to the margins of the prominent inversion structures of the Flechtingen High, the Harz**
**Mountains and the Lusatian-Sudetic High, which brought Permian to Lower Cretaceous sandstones into the erosion level. Slightly modified from Voigt (2015).**

During the Maastrichtian, the Prignitz High was flooded and completely covered with sands of the Nennhausen Formation (Fig. 11). The facies belts show a pattern of extended shallow marine sands which give way to marlstones of mid-shelf
environments. They occur on top of the Prignitz High at the same elevation as on the former marginal troughs. Glauconitic sands of the Nennhausen Formation reach their highest thickness of 600-1000 m in the peripheral sinks of salt diapirs. Together with the thinner deposits, they reflect a single extended coastal facies belt (Ahrens et al., 1965; Voigt, 2008). They are overlain conformably by Paleocene marine deposits. The end of inversion of the western Prignitz High can be dated to have occurred in the late Campanian to early Maastrichtian.


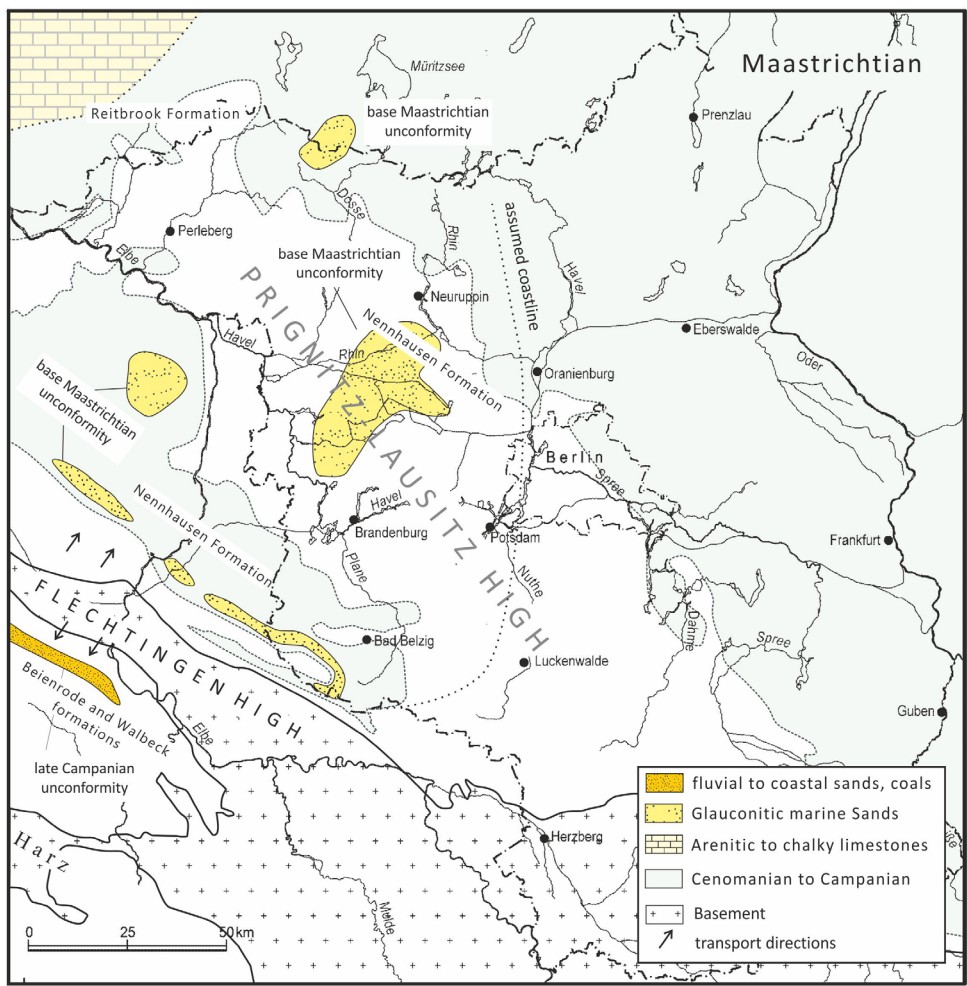

**Fig. 11: The end of inversion of the Prignitz High is shown by its cover of Maastrichtian marine sands (Nennhausen Formation).**
**Maastrichtian deposits extend across the Altmark Basin and to the northern Subhercynian Block (Walbeck Formation). The latter**
**overlies conformably Late Campanian marine deposits (Beienrode Formation), which rest unconformably on Triassic and**

**Jurassic. Post-inversion deposits are mainly restricted to peripheral sinks of salt diapirs and the narrow secondary trough of the Flechtingen High, where they were protected from later erosion. The widespread marine facies belts point to complete flooding of both former highs and basins. Modified from Voigt (2015).**

## 6       Discussion

Basin re-organisation and formation of new depocentres that spatially coincide with those of the Coniacian and Santonian, occurred in the majority of the investigated basins already during the late Cenomanian (96 Ma). This is about five Ma earlier than previously deduced from AFT ages (Fig. 12). In particular, basins clearly related to the compressive reactivation of major normal faults seem to be reflect an early begin of thickness differentiation, like the Osning Thrust and the Rheder Moor-Oythe Thrust bounding the Lower Saxony Basin; and the central part of the Lusatian Thrust, where slices of Jurassic

deposits prove the reactivation of a previous major normal fault (Voigt, 2009). The same is true for the re-activated normal fault of the Quedlinburg Graben within the Subhercynian Basin (Westerhausen Thrust), where its late Cenomanian reverse activity is indicated by strongly reduced thickness and erosional unconformities.

The oldest AFT data point to a rapid passing of basement rocks of the PAZ around 89-90 Ma (Turonian-Coniacian transition) presently exposed at the surface. The precise timing when these rocks entered the PAZ is probably not resolvable

by thermochronology, whereas even small increments (tens of meters) of surface uplift can modify patterns of deposition and erosion. Recent (U–Th)/He ages of the Harz basement, nevertheless, are between 90 and 96 Ma (von Eynatten et al., 2019) and consistent with the results from the basin (fig. 12). The same is true for the Flechtingen High (Fischer et al., 2012) and the oldest ages at the eastern boundary of the Bohemian-Saxonian Cretaceous Basin (Danišík et al., 2010, 2012).

The most precise time constraint from the sediments comes from the Bohemian Cretaceous Basin, where the early to middle

Cenomanian fluvial depositional system is replaced by a marine environment in coincidence with a complete reversal of the main sediment input direction. North- and southeast-directed transport directions of the old northward tilted relief still prevail during the Cenomanian but a source area in the northeast provided the majority of clastic material. The sudden appearance of a northern source area, which coincides with the Lusatian-Westsudetic High that controls the inversion during the entire late Cretaceous (e.g., Tröger, 1965; Skoček and Valečka, 1983; Uličný et al., 2009; Voigt, 2009), strongly hints to

a middle to late Cenomanian begin of inversion.

Wilmsen (2003) reconstructed the facies distribution of the Cenomanian in northern Germany and established a well-funded sedimentary model of Cenomanian hemipelagic and pelagic deposits. High thicknesses of Cenomanian deposits were interpreted as the result of a high-productivity facies belt ("calcisphere system") in contrast to a low productivity belt ("coccolith system") in the north and condensed, accommodation-controlled sections closer to the ancient coast. The belt

with greater thickness, interpreted to be the result of varying sedimentation rates, corresponds to the marginal troughs on both sides of the inverted Lower Saxony basin, proposed here. However, the section with the highest thickness of the model of Wilmsen (2003) is situated in the peripheral sink of the Bokeloh diapir (Wunstorf) and the most condensed section (Langenstein) is situated at the western margin of the Quedlinburg Anticline, close to the Westerhausen thrust, which was

active during Cenomanian (fig. 4). Lines of equal Cenomanian thickness follow the contour of the marginal troughs and not
the orientation of the coast (fig. 2). Highest thickness of the Cenomanian (200 m) occurs in the front of the Rheder-Moor
Oythe Thrust System. Further, the rapid thickness decrease in the South Oldenburg Basin from 200 to 50 m (fig. 9) points
more to enhanced tectonic subsidence than to primary facies differentiation. The influence of salt migration can be excluded
due to low thickness of Zechstein deposits there (fig. 9). Detailed facies investigations of the Cenomanian succession are
necessary to decide, whether subsidence of the marginal troughs or facies differentiation was responsible for thickness
trends.

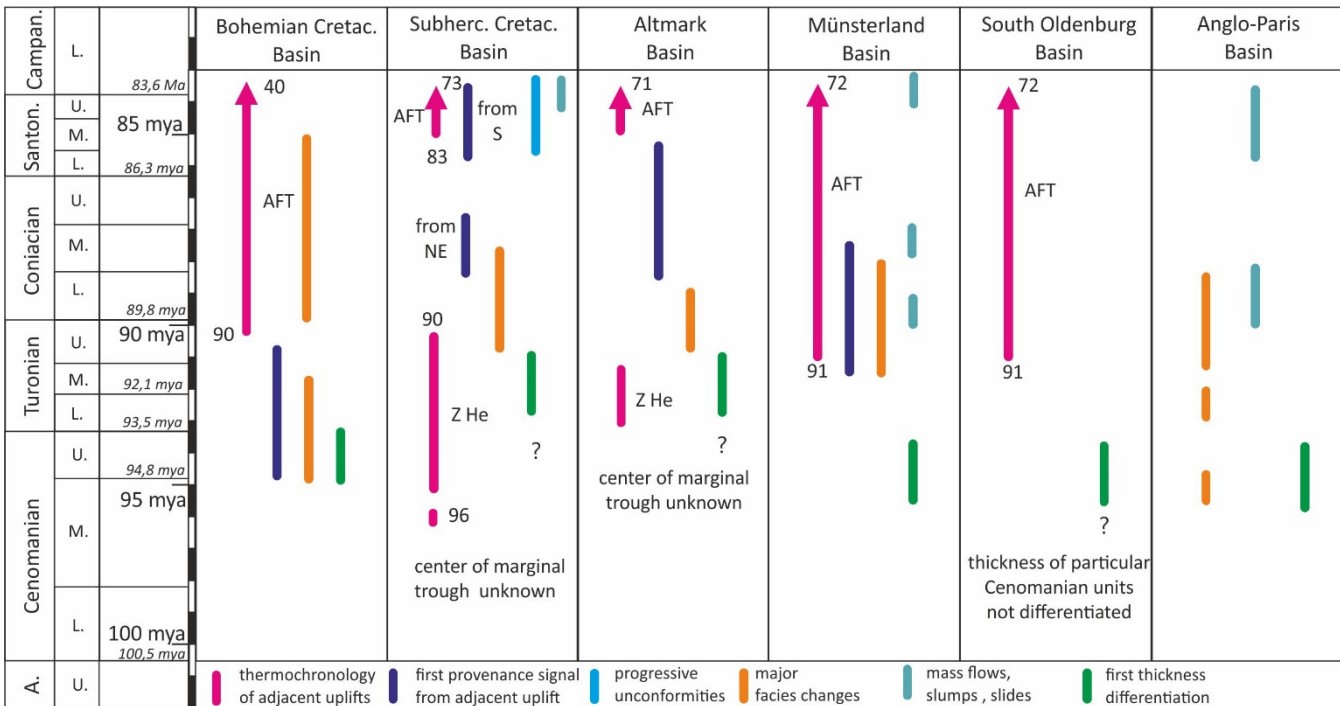

**Fig. 12: Compilation of time constraints fixing the start of inversion. Varying methods basing on thermochronology, sediment redistribution, provenance, progressive unconformities and thickness differentiation show a significant disparity in timing between Cenomanian and Coniacian/Santonian. First evidence of changes in basin configuration occurred already during middle to late Cenomanian times across western and central Europe.**

The positions of the marginal troughs and the principal regions of maximum thickness do not change from Cenomanian to
Turonian (Malkovsky, 1987). Uličný et al. (2009) observed the migration of the sandy margin facies to the southeast from
the Turonian to the Coniacian. They interpreted this pattern to reflect a lateral shift of the source area and concluded that the
basin evolved in a NW-SE-striking transtensional strike-slip system. However, the northwestern margin of shoreface sands
in the Saxonian part of the Bohemian Saxonian Basin migrates about 15 km in the opposite, northwest direction during the

same time (fig. 7). This pattern better matches an increase in size of the uplift than unidirectional displacement of the source area. The increased sediment input beginning in middle- to late Turonian seems to cause a general extension of the sandy facies belt.

The increasing sedimentation rates from about 50 m/Ma in the late Cenomanian to 75-110 m/Ma in the early to middle Turonian and to >300 m/Ma in the late Turonian and lower Coniacian point to accelerated subsidence of the marginal trough accompanied by clastic input which was mostly accommodated by the subsidence close to the uplifting margin. The absence of lower to middle Cenomanian fluvial deposits in the region of highest thickness of Upper Cretaceous sediments and their independent thickness trends prove basin re-organisation and the post-middle Cenomanian formation of the marginal trough. The marginal troughs of the inverted Lower Saxony Basin show only a thickening signal in the Cenomanian close to the uplifting margins but no facies change since the uplifted area did not rise above sea level at this time. As in the Bohemian Saxonian Basin, inversion started slowly and accelerated particularly during the late Turonian and Coniacian. It is still unclear whether this trend comprises the entire Cenomanian or only its latest part, as documented in the well-investigated Bohemian Saxonian Basin.

The Subhercynian Basin clearly shows the influence on sedimentation of a basin-internal structure in the Cenomanian. Cenomanian thickness on the southern flank of the Quedlinburg Anticline is strongly reduced and even an erosional unconformity developed above Middle Cenomanian and below Lower Turonian deposits. The area of this anomaly coincides with the northeast-dipping bounding fault of a half graben filled with Hauterivian and Barremian coastal sandstones. We interpret the reduced thickness to indicate the start of inversion. Alternatively, the uplift could be related to salt tectonics, but the northern flank of the narrow anticline shows no evidence for similar reduction of Cenomanian thickness. Surprisingly, no clear evidence for Cenomanian tectonic activity has yet been observed close to the Harz uplift even though the Harz mountains are the most prominent structure. The widespread erosion below the middle Santonian unconformity at the northern margin of the Harz probably erased the entire sedimentary record of earlier fault activity. A possible thickening of Cenomanian deposits in the axis of the marginal trough is not known, because no borehole reached the Cenomanian in the deeply subsided basin part close to the Harz uplift margin.

In contrast to these examples of early activities in basin evolution, the deep marginal troughs of the Altmark Basin show no significant variations in Cenomanian thickness. Neither the Harznordrand Thrust nor the Gardelegen Thrust are proven to have formed from inherited normal faults (e.g., Voigt et al., 2009; Malz et al., 2020). They may have nucleated later in stronger lithosphere in comparison to the easily re-activated normal faults of the Lower Saxony Basin and the Quedlinburg Anticline in the Subhercynian Basin. No drilling or high-resolution seismic data exist for the deeply subsided centers of the Subhercynian and Altmark basins so that the question of an early start of inversion-related subsidence in these basins remains open.

The early start of inversion tectonics in the Cenomanian is observed in a variety of basins of central Europe, but has not been reported from the inverted Polish and Danish basins until now. Here, a detailed areal evaluation of Cenomanian thicknesses is necessary to date the start of inversion tectonics more precisely or to conclude that only the southern basin exhibits the

early start because deformation propagated northward. An early start of inversion was excluded for the Regensburg Basin south of the Central European Basin on the base of a detailed basin study (Niebuhr et al. 2014). Instead, the Cenomanian transgression show a stepwise flooding of the basement surface towards the later high. Depositional thickness started to rise during the Middle Turonian, accompanied by a significant increase of the mean grainsize. This could indicate that the marginal trough of this basin is not preserved or the uplift started later due to a higher strength of the lithosphere. Because the Anglo-Paris Basin shows also clear evidence of Cenomanian contractional fault activity (Mortimore and Pomerol, 1991), we assume a synchronous onset of inversion in Central and Western Europe with the mentioned exceptions.

The incomplete preservation of marginal troughs in Central Europe only in a few cases allows to constrain the end of inversion more precisely than to the Late Cretaceous to Paleogene (Fig. 13). The regional uplift after basin inversion, which affected both the highs and the adjacent syn-inversion basins, obscured the signal of the change in basin configuration, which occurs with the end of compression. A major erosion surface bevelled both highs and basins. This unconformity appears flat outside the areas of salt migration, but marine deposits of varying age, reaching from the Maastrichtian to late Eocene and early Oligocene, cover it and indicate a long-lasting evolution of a low-relief paleosurface. This surface is very gently inclined towards the northwest and the deposits covering the unconformity are continuously younger to the south.

Primary and secondary marginal troughs are therefore only preserved where an area remained below base-level. The Harz Mountains with the Subhercynian Basin and the Lusatian-Sudetic High with its paired marginal troughs show a time gap of 30-40 Ma between the youngest Santonian/lower Campanian sediments of the marginal trough and the Eocene post-inversion unconformity (Fig. 13).

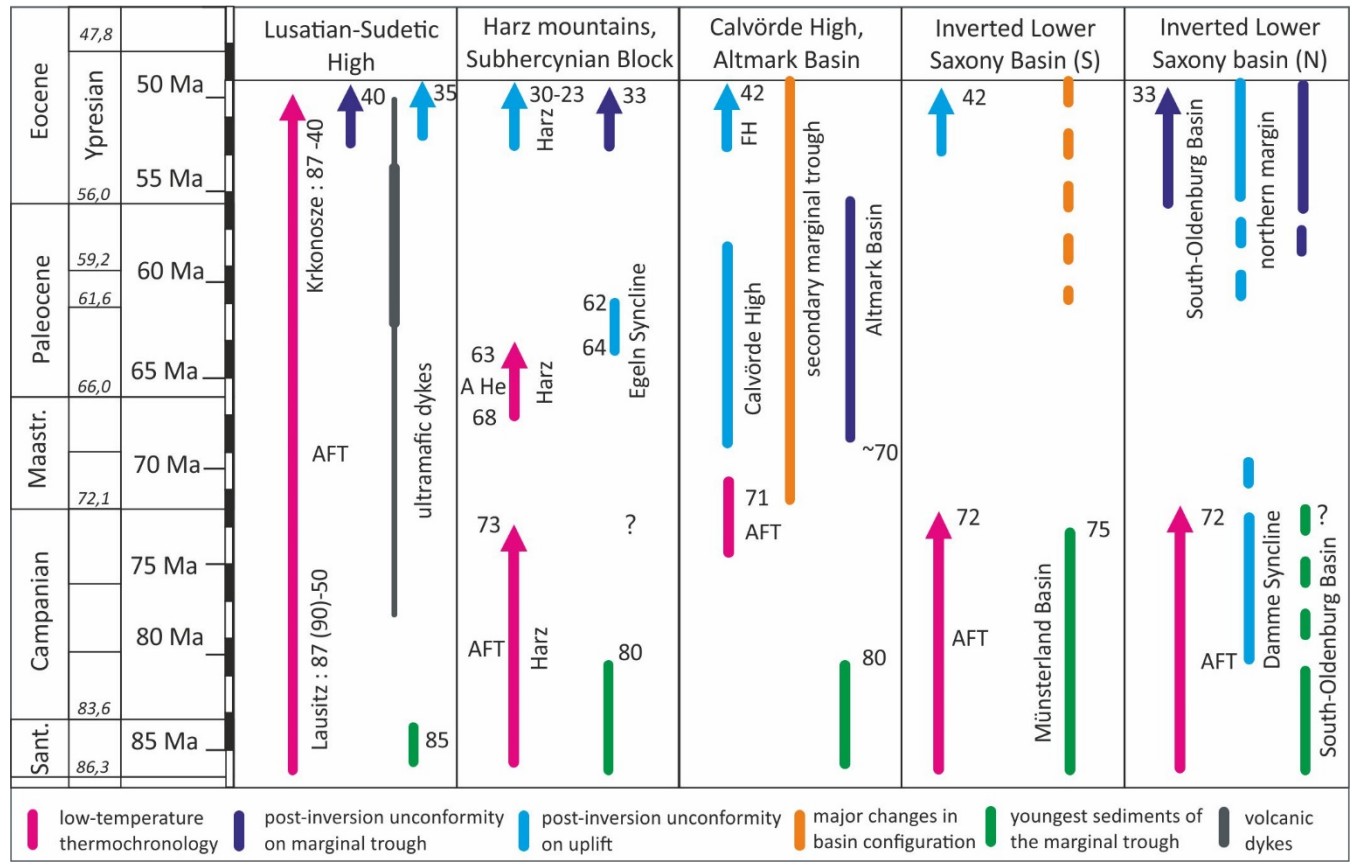

**Fig. 13: Timing of the end of inversion tectonics in Central Europe is difficult to constrain due to a post-inversion uplift of most structures, involving both highs and related marginal troughs. While sedimentary covering of the Lower Saxony Basin with post-inversion deposits occurred already in the Campanian, a significant shift in the evolution of the marginal troughs occurred either in the Maastrichtian or later. FH: Flechtingen High.**

The marginal troughs situated more to the north preserved better evidence of the end of inversion due to the local preservation of Maastrichtian and Paleocene deposits. These indicate a shift of the basin axis between the early Campanian and Maastrichtian times. On top of the inverted highs, the ages of the overlying sequences differ between the Prignitz High (early Maastrichtian) and the inverted Lower Saxony Basin (latest early Campanian) by seven to nine million years. In both cases, marine deposits were preserved. Although Maastrichtian and Paleocene units are generally preserved only in isolated marginal troughs of salt diapirs, marine deposits prove that the surrounding highs had also been close to sea-level and were flooded by the advancing sea. In the case of the Damme syncline on top of the inverted Lower Saxony Basin, the influence of a salt diapir can be excluded.

The diachrony between the basal deposits overlying the unconformity on the Prignitz High, in the Damme Syncline and in the Altmark Basin probably suggest slowly waning uplift and transition to subsidence, in contrast to the sudden shift of the axis of marginal troughs in the Danish Basin within the Danian (early Paleogene). This can probably be explained by

continuing salt migration: In addition to tectonics, the preservation of sediments is controlled by the re-distribution and subsolution of thick Permian (Zechstein) salt, which is or was present in the majority of inversion-related basins in Central Europe, except the Bohemian Saxonian Basin and the Regensburg Basin.

In comparison to the well-investigated primary and secondary marginal troughs of the Danish basin and to the deeply buried inversion structures in the North Sea and western Europe, the timing of the end of late Cretaceous inversion in Central

Europe remains poorly constrained. The proposed secondary marginal trough of the Altmark Basin predates the Danish secondary marginal trough, is closer to the previously active thrust and is deeper than the Danish example. The start of subsidence in the Maastrichtian, which lasted until the late Eocene, could also be the result of salt migration into the adjacent pillows north of the basin due to doming of the southern region. Despite this possible alternative interpretation of the secondary marginal trough formation, the unconformity cuts both the Altmark Basin and the intrabasin anticlines formed

between early Campanian and Maastrichtian times and thus marks the end of differentiated subsidence. This is in contrast to the exact timing of the shift of marginal troughs in the Danish basin in the Paleocene, interpreted as the transition from convergence to relaxation by Nielsen et al. (2005) and Nielsen et al. (2007). A base-Maastrichtian unconformity was also observed in the Vlieland Basin (Deckers and van der Voet, 2018). Indications of a change in the stress field at the Campanian-Maastrichtian transition were found in palaeo-stress records in the Mons Basin (Vandycke and Bergerat, 1989).

The onset of basin inversion in Central Europe coincides with major changes in relative plate motion between Africa, Iberia and Eurasia (Kley and Voigt, 2008; Rosenbaum et al., 2002; Seton et al., 2012), which in turn coincide with a mid-Cretaceous global plate reorganization event (Scotese et al., 1988; Veevers, 2000; Matthews et al., 2012). The exact age of this event is difficult to pinpoint because it occurred in the magnetic quiet period or Cretaceous Normal Superchron (CNS) between chrons M0 and 34 (120.4 – 83.5 Ma). Matthews et al. (2012) used interpolation between magnetic anomalies to

suggest that the reorganization took place between 105-100 Ma, in Albian to Cenomanian times. The start of deformation in Central Europe is contemporaneous with the global Late Cenomanian sea-level rise, a consequence of the rapid formation of new oceanic crust. This supports a causal link between global plate reorganization and intraplate deformation. The termination of inversion was either due to a latest Maastrichtian to Paleocene drop in Africa-Iberia-Europe plate convergence (found by Rosenbaum et al. (2002), but not Vissers and Meijer (2012)) or to mechanical weakening of the Iberia-Europe

plate boundary caused by incorporation of continental crust (Dielforder et al., 2019).

## 7     Conclusions

The start of inversion in many Late Cretaceous basins of Central Europe can be dated to about 96 Ma, 5 Ma earlier than hitherto assumed based on detailed analysis of new depocenters forming close to the inverting structures. The first signals of

inversion are weak, because the sedimentation rates are only approximately 20% of the maximum sedimentation rates attained during the Coniacian and Santonian. This slow increase in sedimentation rate probably also indicates a slow

development of the tectonic loads inducing the subsidence. There is no evidence for a different compression direction during the Cenomanian in the Central European Basin, except in the Saxonian part of the Bohemian-Saxonian Basin, where a secondary oblique axis of a single depocenter develops and southeast-trending highs appear, not obviously correlated to the previous paleodrainage system, which reflects in general a dissected peneplain gently tilted to the north.

A regional unconformity between the inversion structures (highs and basins) and a covering sequence developed between the early Campanian and Maastrichtian. A similar surface was formed again during the Maastrichtian and Paleocene, probably due to erosive levelling in the course of recurrent transgressions, and resulted neither in major deposition nor major erosion. Only in the surroundings of active diapirs and in a few secondary marginal troughs, relict sediments of coastal plains and shallow marine environments witness the existence of an extended marine cover on this surface. The covering of inverted structures by deposits varying in age from Campanian to Maastrichtian indicates rather a gradual deceleration than a sudden end of compression and uplift. Large-scale salt migration is probably the main reason for preservation of a complete marine succession from Maastrichtian to Paleocene on top of the Prignitz High, which gives strong evidence that the inversion in this part of the basin was finished before the late Maastrichtian. The end of basin inversion should be better constrained in the areas deep in the subsurface (northern slope of the inverted Mid-Polish Trough, Grimmen Swell, Danish Trough), because the conservation of the unconformity is much better there than in the south, where several transgressions wore down the original surface.

The start of deformation in Central Europe was probably caused by a global plate reorganization event. This event induced both the changes in plate kinematics and the coeval global late Cenomanian sea-level rise due to a peak in the production of new ocean floor. The reorganization event, dated to Albian to Cenomanian time between 105-100 Ma, would be the earliest possible age for the onset of compression and basin inversion.

*Author contributions.* TV: general idea, conceptualization, borehole data acquisition, thickness maps and interpretations, drawing of Figs. 3-13, writing of the original draft, funding acquisition (EU, DFG). JK: conceptualization, drawing of Fig. 1, discussion, additional and critical information, validation, critical review in the pre-publication stage, funding acquisition (DFG). SV: contributions on the general problem of Late Cretaceous basin inversion in Central Europe, data acquisition of Cenomanian thicknesses in the northern study area, compilation of thickness data, drawing of Fig. 2., discussion and validation, revision, critical review in the pre-publication stage.

*Competing interests.* The authors declare that they have no conflict of interest.

**Acknowledgements**

This paper was initiated by discussions with the late F. Kockel, the "godfather of Central European Basin inversion" already 20 years ago. We gratefully acknowledge his inspiration. The Geological Surveys of Brandenburg, Saxony and Saxony Anhalt supported this work by providing access to research borehole data and some cores for new interpretations. We thank

especially Frank Horna, Gerhard Beutler, Karl-Heinz Friedel, Bodo Ehling, Werner Stackebrandt and Michael Goethel for granting us access to old cores and the data repositories and helpful hints concerning rare data and maps.

We thank Roland Nádaskay, Jiří Adamovic and Markus Wilmsen for discussion of facies and thickness in the Bohemian Saxonian Basin. They contributed also with critical remarks and additional information which improved the quality of the paper. Especially, we thank the two reviewers Jef Deckers and Pawel Aleksandrowski for their insightful and constructive remarks which improved the quality of the manuscript. Markus Wilmsen and Roland Nádaskay contributed with ideas and critical remarks during the review process. Christoph Heubeck is gratefully acknowledged for polishing the language in a late stage of the paper.

This work was in part funded by German Research Foundation grants KL 495/9, GA 457/6 and the SPP 1135: "Dynamics of sedimentary basins". The European Union financially supported research in the Bohemian-Saxonian Cretaceous Basin in the frame of the projects GRACE and ResiBil. The compilation of borehole data in these projects sparked the idea that the inversion of the Lusatian-Sudetic High started already in the late Cenomanian.

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
