# Peer review of "Dawn and Dusk of Late Cretaceous Basin Inversion in Central Europe"

_Solid Earth, 2020_

## Referee Comment (RC1) · Pawel Aleksandrowski (Referee) · 31 Dec 2020

General comments

The manuscript of "Dawn and dask of Late Cretaceous basin inversion in Central Europe" promises a briliant, highly interesting and useful review paper, which also brings new solutions following from the reviewed material. These new solutions concern, among others, finding that a rather synchronous beginning of basin inversion in central Europe occurred ∼5 million years earlier than generally thought so far and showing the complexity of the problem of how long the inversion lasted, when it finished and how one can look for the right answer to it. After a highly informative exposition of the questions related to the Late Cretaceous tectonic deformation over a vast area of cen-

tral to west-central Europe, the methods used to uravel the questions of its timing are presented and shortly discussed. These include stratigraphical analysis of reflection seismic sections, thermochronological data (AFT and AHe), thermal maturity of sediments, sediment composition and facies distribution. Subsequently, attempts at timing the deformation in a number of inverted basins are made, from the Lower Saxony basin in the NW, to basins adjacent to the Lausitz-Karkonosze high in the SE, using the results obtained with the above methods, and specific features of each of these basins are shown. In the discussion the results from the reviewed basins are correlated and compared and the conclusions are drawn, concerning the roughly synchronous but slow commencement of the inversion already in Cenomanian, a gradual deceleration of the compression and uplift in Campanian to Mastrichtian times and the difficulties in precise timing the very end of inversion, which might have happened diachronously in various basins. A humorous complaint of mine is, however, that the paper, while discussing the conditions and circumstances of the Late Cretaceous inversion in central Europe, leaves unanswered my favourite question if there were any real mountain chains at that time throughout the area. The authors, though not openly, seem, however, to believe that the erosion was in general effective enough as to keep up with the uplift, so not much hope remains as to the existence of a Late Cretaceous mountainous landscape over the so called Paleozoic platform of central Europe.

Another remark concerns an apparent missing in the paper of a recent concept of "deep" burial of the NE Bohemian Massif, which is now believed to have affected most part of the Sudetes during the Late Cretaceous. The concept is based on new low-temperature thermochronological results. This issue was first mentioned by Danišik et al. (2010, 2012) and later claimed by Sobczyk et al. (2015, 2020). Also Botor et al. (2019) found similar data for the Intra-Sudetic Basin rocks. In the paper by Sobczyk et al. (2020) there is a discussion about possible time frames for the Cretaceous basin evolution in the Sudetes, which was based not only on data from basement rock samples but, importantly, on those from sedimentary rocks. It would, perhaps, be of some use to look at these results in the context of scenarios you propose in this paper (the

links to the above mentioned papers are given in the end of this review).

In spite of the above possible minor shortages, the manuscript fully deserves being published in Solid Earth, though it also deserves a number of minor improvements to be introduced or some minor issues rethought, all of which are listed below as specific comments and technical corrections.

In particular, referring to the SE checklist for reviewers, I should state that: (1) The paper addresses scientifically important questions of wide, international interest, lying within the scope of SE. (2) It presents an original review of and a thorough discussion of the existing data and applicability of a range of methods to solve a scientifically significant problem of dating the Late Cretaceous basin inversion that occurred over a vast area of Central Europe and whose results are still of much importance in the regional geology. The reviewed results add important new elements to the knowledge on the studied object. (3) The presented approach, assumptions made, methods selected and results discussed are appropriate to achieve the goals set up by the authors. (4) The presented interpretation and conclusions are justified by the data employed and the presented ways of reasoning seem to be correct. (5) The paper gives proper credit to the related work by earlier authors (though some minor completion is suggested in specific comments) and its original contribution lies in correlation of relatively large amount of the existing data of various character, their critical discussion and, on this basis, arriving at novel conclusions. (6) The title is perfectly fitting the paper's content. (7) The abstract is sufficiently summarizing the paper's content, though, in my opinion it might contain slightly more information on the paper's conclusions. (8) The overall presentation is well structured, clear and linguistically well written. (9) There is no need to significantly change any part of the manuscript; hints how to improve some minor issues in the text or figures are given in the specific comments. (10) The amount and quality of referenced work seems appropriate.

Specific comments:

Line 42: the "Lausitz-Krkonosze High" is – as concerns its location - only a part of the Late Cenozoic uplifted "Sudetic block", north of which there is "Fore-Sudetic block" – in the present-day Polish geological nomenclature. Both have Palaeozoic and older basement below the Cenozoic, but the Sudetic block is uplifted and the Fore-Sudetic one - downthrown. These two units are identified in Fig. 5 as the Lausitz-Krkonosze High and Northsudetic High, respectively. Changing the nomenclature from the "Lausitz-Krkonosze High" and "Northsudetic High" into the "Lusatian-Sudetic high" and "Fore-Sudetic high" seems to be worth considering.

Line 141:- Consulting the paper of Sobczyk et al. (2020) is suggested for AFT data from sedimentary rocks, which complete those coming from the basement and are discussed there in this context.

Lines 365-368: It is disputable whether the geographical names applied to creating names of geological structures/units should be in original national languages used (to-day?) on their location or in their English (or anglicised version). A good example is provided by the "Lausitz Thrust". It occurs in Germany and Czechia and in the latter country is named "Lužicke nasunuti". The English (← Latin) name for Lausitz (=Czech and Polish "Lužice") is "Lusatia", so maybe the "Lusatian thrust" (already functioning in English-language papers by Polish and Czech authors) might be a better choice? Such a solution may apply to some other names in the paper. (By the way, I see now, in line 593 the "Lusatian block", which means that this solution is, actually, already applied in the paper in some cases. So, maybe, "only one way of applying names" would be beneficial for the consistency of the editorial aspect of the paper? Moreover in the paper's text, here and there one can see a tendency, which I personally prefer, to start the common-name parts of geological names with lowercase letters, which is, btw, not observed in Fig. 5 (Graben, Basin, High, Deformation Front) and which should be correlated with the spelling elsewhere in the text.

Line 370: I would suggest taking a look also at the Intra Sudetic Basin and the Nysa Graben (not labelled on the map in your Fig. 5, but defining an irregular NNW-SSE

directed lense within the basement, to the E of the Bohemian-Saxonian-Cretaceous basin and below the Northsudetic high label) as this structure contains quite an interesting Cretaceous stuff (see Botor et al., 2019, and Sobczyk et al., 2020).

Lines 465-466: This is debatable as the age of the peneplains is actually, still unknown - more likely this is just one of the possible scenarios – see Danišik et al. (2010) and discussion therein.

Line 645: Having a look on a discussion about the influence of a thrust regime on remodelling the Nysa Kłodzka Graben (Sobczyk et al. 2020) may be useful.

Line 657: Again, the paper by Sobczyk et al. (2020) can be of interest in this context, as it contains a relatively detailed discussion on the inversion onset in the Sudetes, based on data coming from both the basement and sedimentary cover rocks.

Fig 1: On the Mesozoic-Cenozoic tectonic map of Central Europe "the main thrusts/reverse faults" of presumed mainly Late Cretaceous age that are marked with heavy barbed lines along the Tornquist-Teissyere zone and the Polish trough are – to my knowledge – not known from the available seismic data as major, long distance reverse faults in the Permo-Mesozoic fill of the Polish part of the Central European basin (Polish basin). On the other hand, the similarly marked major "thrusts", along the NE margin of the Bohemian Massif (the Sudetes' Boundary Fault and the Middle Odra Fault) and of the Bruno-Vistulian Block are all – according to the available data – very steep fractures of original strike-slip origin. Due to their near verticality, it is difficult to term them "thrusts" or even "reverse faults", with the notable exception of the Lusatian (Lausitz) thrust to the north of the Bohemian-Saxonian Cretaceous Basin, which is, indeed, well exposed and clearly verified as a major thrust (or rather reverse fault – due to its high-angle attitude). The above remarks are, nevertheless, of minor general importance and can be disregarded.

Fig. 4. The significance of steep hachure lines on the N margin of the Harz Mts eroded area remains enigmatic and needs explanation.

Fig. 5. In my opinion the "Lausitz-Krkonosze High" might be preferably (here and in the paper's text) replaced by "only English"- "Lusatian – Sudetic high" (or still "Lusatia – Sudetes high"). At the same time, the "Northsudetic high" –would be better replaced by "Fore-Sudetic high" (as the term "Fore-Sudetic block" – in contrast to the, now uplifted, "Sudetic block" - is widely applied to this area in the tectonic literature).

Since such important Cenozoic tectonic elements as the Alpine deformation front are included in the map, I suggest considering usage of a broken line to mark the position of the Sudetic Boundary (Marginal) fault, which definitely existed in Late Cretaceous times, as a remnant Late Variscan strike-slip fault. Its possible importance would lie in supplying a reader of Fig 5 with a reference structure very well known from the present-day geology, according to which he/she will be able to better confront the map of Fig. 5 with "normal" geological maps of the area.

In terms of the lithologies included in the legend, the Opole basin should be filled with marl and not chalk and the Cretaceous of the Intra-Sudetic basin (not labelled as such on the map, but defining an irregular NNW-SSE directed lense inside the basement rocks, to the E of the Bohemian-Saxonian-Cretaceous basin and below the Northsudetic high label) should combine marl and sand.

The river Oder (Polish or Czech – Odra) should have its course significantly extended upstream, well beyond the frame of Fig. 6, since it is altogether strangely abandoned on the map still west of Wroclaw. It is also some of Oder's main tributaries (such as the Lausitzer Neisse and Glatzer Neisse) that should be added to the map (maybe also accompanied by the present-day state frontiers) to make it easier to the reader to find where "he/she is" geographically on the map.

Figs 10 and 11 have no explanation in their legends for the lithological(?) division covered with crosses.

Fig.12: Possibly it would be useful to show separately thermochronological signals reported from the basement and those coming from sedimentary rocks, especially for

the Cretaceous Bohemian Basin, for which such data are reported in the literature. Another suggestion is a recommendation to check thermal modelling results (instead of using only 'raw' ages) as this might inspire more detailed basin history reconstructions.

Technical corrections:

Line 27: North See should be transfomed into North Sea.

Line 85: "where" should be replaced by "were".

Line 88: Shouldn't "Wernigeröde" be replaced by "Wernigerode"?

Line 386: The citation of "T. Voigt, 2009" should probably read "T. Voigt et al., 2009", since the latter item has the closest shape to "T. Voight, 2009" on the reference list.

Line 427: Shouldn't the "northwestern edge of the Lausitz-Krkonosze high" be rather the "southwestern......." one?

Line 607: a reference to Kaeßner et al. 1999 is not reflected on the list of references. Fig. 2: The contours of the present-day European coastline should preferentially be better visible.

References cited

Danišik et al. 2010 https://doi.org/10.1016/j.geomorph.2009.11.010

Danišik et al. 2012 http://dx.doi.org/10.1029/2011TC003012.

Sobczyk et al. 2015 https://doi.org/10.1016/J.TECTO.2015.02.021

Botor et al. 2019 https://doi.org/10.1007/s00531-019-01777-9

Sobczyk et al. 2020 https://doi.org/10.1111/TER.12449

---

## Referee Comment (RC2) · Jef Deckers (Referee) · 12 Jan 2021

Dear Editor,

It is a very valuable work that adds a lot of knowledge on the inversions in central Europe. The article is well written and provided with nice illustrations.

My main comment is that more literature from the surroundings has to be considered for establishing the timing/dynamics of inversion pulses or phases. I see very few citations of literature from the Netherlands and Belgium, while in these regions several papers were recently written on the dusk of inversion which could/should be integrated. In fact the authors struggle with pinpointing the dusk of inversion(s), so the article could strongly benefit from knowledge on this topic from the surrounding areas. Instead

of discussing the dawn of inversion based on solely the results of their study area, it would be better if the authors compared the timing with those established in the surroundings in order to either confirm or contradict the existing literature on that. The authors also found several indications for Late Cretaceous inversion pulses (distinct Santonian-Campanian and Maastrichtian phases?) which could be highlighted more in the paper. Also on this possible distinct Maastrichtian pulse, literature exists from the Netherlands and Belgium.

Another important comment is that the article would benefit from a figure that provides a general overview on the stratigraphy in the different troughs or inverted structures.

Please find my comments on the text and figures in the PDF in attachment.

Best regards, Jef Deckers

Please also note the supplement to this comment:
https://se.copernicus.org/preprints/se-2020-188/se-2020-188-RC2-supplement.pdf

**Supplement:**

[revised manuscript text omitted]

---

## Author Comment (AC1) · 16 Mar 2021

Dawn and Dusk of Late Cretaceous Basin Inversion in Central Europe T. Voigt, J. Kley, S. Voigt Comments to the review of Pawel Aleksandrowski Thank you for the careful and comprehensive review. We added some comments and changed also the names as you proposed. Thanks for the hints to additional literature from Poland. As we started to write the paper two years ago, we missed some new interesting data.

We want to answer some of you questions: 1. …...leaves unanswered my favourite question if there were any real mountain chains at that time throughout the area.

The answer is given by the grain size of the marginal sediments. At the Lusatian thrust

in Saxony, you can find clasts within the Turonian up to 10 cm (quartz, ironstones), maximum 15 cm (reworked sandstones). At the Harz margin, in the Santonian: 6-8 cm (Buntsandstein), in the Campanian 2-10 cm (Lower Carboniferous and Devonian slates). Other conglomerates more or less negligible. The rivers of the Harz and the Karkonosze transport granitic clasts up to 0.5 m into the foreland. The morphology of the Cretaceous mountains was hilly. Uplift rates were too small and erosion rates too high to create steep rivers and high cliffs at the coast.

2. …..an apparent missing in the paper of a recent concept of "deep" burial of the NE Bohemian Massif, which is now believed to have affected most part of the Sudetes during the Late Cretaceous

Yes, we did not mention it, because these uplifts occurred after the Late Cretaceous inversion. At the same time as Danišik and Sobczyk published their results, we made some still unpublished investigations of the thermal maturity of some coals and drift wood pieces in northern Bohemia and in Saxony. We have the same results: 3-4 km of burial! As the basement rocks do not show a complete reset (Käßner et al. 2020), the maximum Cretaceous burial depth is also 3-4 km in the BCB. This considerable post-depositional uplift occurred in the period between the Cretaceous and the Oligocene. The driving force was probably different, considering timing and orientation of uplifts (escape tectonics of the Alps as driving force for the origin of the Carpathians?). We added this amazing facts to the paper.

Line 42: the "Lausitz-Krkonosze High" is – as concerns its location - only a part of the Late Cenozoic uplifted "Sudetic block", north of which there is "Fore-Sudetic block" – in the present-day Polish geological nomenclature. Both have Palaeozoic and older basement below the Cenozoic, but the Sudetic block is uplifted and the Fore-Sudetic one - downthrown. These two units are identified in Fig. 5 as the Lausitz-Krkonosze High and Northsudetic High, respectively. Changing the nomenclature from the "Lausitz-Krkonosze High" and "Northsudetic High" into the "Lusatian-Sudetic high" and "Fore-Sudetic high" seems to be worth considering.

You are right the nomenclature is not easy. It starts with the Krkonoše – Riesengebirge – Karkonosze and does not end with Lausitz – ŁuÅijyce – Lužice. Lusatian-Sudetic High is the much better and more comprehensive term, because the Lausitzer Gebirge, Góry Kaczawskie, Jizerske hory, Góry Sowie and some other structures were involved in the uplift too. We used the term Lusatian-Sudetic High throughout the paper as you proposed.

Line 141: Consulting the paper of Sobczyk et al. (2020) is suggested for AFT data from sedimentary rocks, which complete those coming from the basement and are discussed there in this context.

We added a short section, although we think that the enormous post-Cretaceous inversion reset does not fell into the Subhercynian phase, but has to be considered as a different event. Instead, the heating shows that the eastern Sudetes were initially part of the marginal trough. Nevertheles, it is simply not in agreement with adjacent areas of the basin, to deposit more than 4 km in the short period of 5 Ma between Late Cenomanian and Turonian (The section is still preserved in the Innersudetic syncline near Adršpach/Broumov and Gór Stołowych with limited thickness in line with other parts of the Bohemian-Saxonian Cretaceous Basin). A later (Maastrichtian to Paleogene) regional uplift is assumed from our side.

Lines 365-368: It is disputable whether the geographical names applied to creating names of geological structures/units should be in original national languages used (today?) on their location or in their English (or anglicised version). A good example is provided by the "Lausitz Thrust". It occurs in Germany and Czechia and in the latter country is named "Lužicke nasunuti". The English (Latin) name for Lausitz (=Czech and Polish "Lužice") is "Lusatia", so maybe the "Lusatian thrust" (already functioning in English-language papers by Polish and Czech authors) might be a better choice? Such a solution may apply to some other names in the paper. (By the way, I see now, in line 593 the "Lusatian block", which means that this solution is, actually, already applied in the paper in some cases. So, maybe, "only one way of applying names"

would be beneficial for the consistency of the editorial aspect of the paper? Moreover in the paper's text, here and there one can see a tendency, which I personally prefer, to start the common-name parts of geological names with lowercase letters, which is, btw, not observed in Fig. 5 (Graben, Basin, High, Deformation Front) and which should be correlated with the spelling elsewhere in the text. The names were changed as proposed.

Line 370: I would suggest taking a look also at the Intra Sudetic Basin and the Nysa Graben (not labelled on the map in your Fig. 5, but defining an irregular NNW-SSE directed lense within the basement, to the E of the Bohemian-Saxonian-Cretaceous basin and below the Northsudetic high label) as this structure contains quite an interesting Cretaceous stuff (see Botor et al., 2019, and Sobczyk et al., 2020).?? The names of the structures were added. A problem is still the Opole Basin, because we. used a nice polish overview map (without Cenozoic cover) from 1972. I recently saw the very detailed map of Irek Walaczsyk that the structure is much more complicated. It would be possible to change it, but it would cost much time.

Lines 465-466: This is debatable as the age of the peneplains is actually, still unknown - more likely this is just one of the possible scenarios – see Danišik et al. (2010) and discussion therein. Section was changed

Line 645: Having a look on a discussion about the influence of a thrust regime on remodelling the Nysa Kłodzka Graben (Sobczyk et al. 2020) may be useful.

Yes, We know this work, but did not consider it as the main uplift occurred after inversion. But we added a short section. This seems a really interesting work to find out, what caused this enormous young uplift (emplacement of the Carpathian nappes from the southwest?)

Line 657: Again, the paper by Sobczyk et al. (2020) can be of interest in this context, as it contains a relatively detailed discussion on the inversion onset in the Sudetes, based on data coming from both the basement and sedimentary cover rocks.

This was considered in the text. It is really a very interesting result, because I (T. Voigt) assumed the thick cover already in the nineties (diagenesis of sandstones and claystones, formation of dickite as a marker of high temperatures in the porespace and high mature coals in the Cenomanian in Saxony, but was not able to fix it. Recently we had a masters thesis on the thermal maturity of the Cretaceous rocks – the results were the same as in the Intrasudetic basin. We wrote an additional section.

Fig. 1: On the Mesozoic-Cenozoic tectonic map of Central Europe "the main thrusts/reverse faults" of presumed mainly Late Cretaceous age that are marked with heavy barbed lines along the Tornquist-Teissyere zone and the Polish trough are – to my knowledge – not known from the available seismic data as major, long distance reverse faults in the Permo-Mesozoic fill of the Polish part of the Central European basin (Polish basin). On the other hand, the similarly marked major "thrusts", along the NE margin of the Bohemian Massif (the Sudetes' Boundary Fault and the Middle Odra Fault) and of the Bruno-Vistulian Block are all – according to the available data – very steep fractures of original strike-slip origin. Due to their near verticality, it is difficult to term them "thrusts" or even "reverse faults", with the notable exception of the Lusatian (Lausitz) thrust to the north of the Bohemian-Saxonian Cretaceous Basin, which is, indeed, well exposed and clearly verified as a major thrust (or rather reverse fault – due to its high-angle attitude). The above remarks are, nevertheless, of minor general importance and can be disregarded.

The faults were taken from Ziegler (1990). They should exist because they mark the boundary of the Jurassic grabens and the marginal troughs. To realize the uplift, faults should be at least in the basement. Most of the faults are poorly known, if they are exposed or drilled (several times in Germany), they mostly show angles between 30 to 60° (maximum). Concerning the Lausitz fault: Thrust is the right expression. The observed angles at the good outcrops in Saxony are between 15° and 27° (quarry in Weinböhla, gas pipeline south of Dresden, outcrops near Hohnstein, exploration close to Hermsdorf, mapping in the Zittau mountains). Also in the Czech part, the angle is

much shallower than displayed in the maps (recent small-scale seismic investigations, Czech-German RESIBIL EU-project). Even the Harz-Nordrand-Fault has an angle between 30-40° (Franzke et al 2004).

Fig 4: The significance of steep hachure lines on the N margin of the Harz Mts eroded area remains enigmatic and needs explanation. The hachured line is explained in the text.

Fig. 5: In my opinion the "Lausitz-Krkonosze High" might be preferably (here and in the paper's text) replaced by "only English"- "Lusatian – Sudetic high" (or still "Lusatia – Sudetes high"). At the same time, the "Northsudetic high" –would be better replaced by "Fore-Sudetic high" (as the term "Fore-Sudetic block" – in contrast to the, now uplifted, "Sudetic block" - is widely applied to this area in the tectonic literature).

The terminology has been changed in text and Fig. 5.

Since such important Cenozoic tectonic elements as the Alpine deformation front are included in the map, I suggest considering usage of a broken line to mark the position of the Sudetic Boundary (Marginal) fault, which definitely existed in Late Cretaceous times, as a remnant Late Variscan strike-slip fault. Its possible importance would lie in supplying a reader of Fig 5 with a reference structure very well known from the present-day geology, according to which he/she will be able to better confront the map of Fig. 5 with "normal" geological maps of the area. We added some faults and hope that we meet your wishes.

In terms of the lithologies included in the legend, the Opole basin should be filled with marl and not chalk and the Cretaceous of the Intra-Sudetic basin (not labelled as such on the map, but defining an irregular NNW-SSE directed lense inside the basement rocks, to the E of the Bohemian-Saxonian-Cretaceous basin and below the Northsudetic high label) should combine marl and sand.

We modified the lithology in the Intrasudetic Basin and labelled it.

The river Oder (Polish or Czech – Odra) should have its course significantly extended upstream, well beyond the frame of Fig. 6, since it is altogether strangely abandoned on the map still west of Wroclaw. It is also some of Oder's main tributaries (such as the Lausitzer Neisse and Glatzer Neisse) that should be added to the map (maybe also accompanied by the present-day state frontiers) to make it easier to the reader to find where "he/she is" geographically on the map.

Sorry, the southern course of the Odra river was hidden behind the geology – repaired! The borders were not added as additional rivers, because this map is only made to show the general picture of Cretaceous basins and the relationship to the uplifts.

Figs. 10-11 have no explanation in their legends for the lithological(?) division covered with crosses.

Fixed, the basement was added in the legend.

Fig. 12: Possibly it would be useful to show separately thermochronological signals reported from the basement and those coming from sedimentary rocks, especially for C6 the Cretaceous Bohemian Basin, for which such data are reported in the literature. Another suggestion is a recommendation to check thermal modelling results (instead of using only 'raw' ages) as this might inspire more detailed basin history reconstructions.

All AFT-data were derived from the basement, no data from sedimentary rocks are presented (except of Permo-Carboniferous sediments, because all strata below base Zechstein is considered as basement here). Fission Track ages of the sediments show only the ages of the basement cooling so far, with the exception in Poland and possibly the BCB. The Paleogene intrabasinal uplifts of the BCB are not considered, because their uplift inverted the inversion-related basin and is another story. The mentioned mistakes in spelling in Lines 27, 85, and 88 were corrected.

Line 386: The citation of "T. Voigt, 2009" should probably read "T. Voigt et al., 2009", since the latter item has the closest shape to "T. Voigt, 2009" on the reference list.

The citation T. Voigt, 2009 is correct, it is also replaced in the reference list.

Line 427: Shouldn't the "northwestern edge of the Lausitz-Krkonosze high" be rather the "southwestern......." one? Northwestern is correct; to make it clearer, the assumed margin was added.

Line 607: a reference to Kaeßner et al. 1999 is not reflected on the list of references.

It is Käßner et al. 2020, and is corrected

Fig. 2: The contours of the present-day European coastline should preferentially be better visible.

It was corrected.

We addedd the figures which were corrected.

Please also note the supplement to this comment:
https://se.copernicus.org/preprints/se-2020-188/se-2020-188-AC1-supplement.pdf

———————————————————

[Figure]

**Fig. 1.**

![Map of the North German Basin region showing Cenomanian thickness. Labels include Fennoscandia, North Sea Basin, North German Basin, South-Oldenburg Basin, Inverted Lower Saxony Basin, Münsterland Basin, Subhercynian Basin, Rhenish Massif, Bohemian Massif, North-Sudetic Basin, Bohemian Saxonian Basin, W Sudetic I. Legend: faults, shoreline, coastal sands, land area. Cenomanian thickness: >100 m, 70–100 m, 50–70 m, 30–50 m, <30 m. Scale bar 100 km.]

**Fig. 2.**

[Figure]

[Figure]

**Fig. 3.**

[Figure]

**Fig. 4.**

[Figure]

**Fig. 5.**

---

## Author Comment (AC2) · 16 Mar 2021

Dawn and Dusk of Late Cretaceous Basin Inversion in Central Europe T. Voigt, J. Kley, S. Voigt Comments to the review of Jef Deckers Thank you for the careful and comprehensive review! We corrected the mistakes you pointed out and discussed additional arguments where we stuck to our text. Overall, the review helped us to make some things clearer and more precise. Sorry for the missing literature about Belgium and the Netherlands about Cretaceous inversion; we included these papers, although the focus was initially to the German basins. We have added a short remark on the Paleogene events, but keep our focus on the compressive late Cretaceous event. Our replies will discuss all comments in the original manuscript. We refer to the line

numbers there. 1. Title: As we consider only processes related to the late Cretaceous Basin compression and not later tectonic events, which are oriented in different directions, we will use the present title. The open question is, where the boundary between western and central Europe is situated (our focus was on Germany, western Poland and the Czech Republic. But, as we sometimes refer to the Anglo-Paris Basin, we will include the Roer Valley Graben, the inverted Central Netherlands Basin and the inversion structures in the North Sea additionally during discussion. 2. Line 8: although we worked mainly in the type locality of the "Subhercynian Phase", we avoid the use of this term – the name is too closely related to the locality where the exact timing cannot be fixed. The same is true in our opinion for the Laramide and the Pyreneen phases (established by Stille in the last century). These "phases" suggest worldwide deformation but represent in fact very different processes (as you proved in your own papers) and are not necessarily connected to each other. 3. Line 12: Added (as proposed): on the basis of borehole data and facies and thickness maps 4. Line 15: this is no contradiction, because the end-Maastrichtian timing is often fixed in the sea-covered northern parts, but in the large area of inversion structures in Northern Germany either already Campanian or Maastrichtian transgresses on inverted structures, and no clear change in basin configuration occurs in the Paleogene/Eocene. This is only possible if the uplift rates decline slowly in a time-span of at least 5-10 Ma, regardless which processes are responsible for this. 5. Line 18: again, the boundary between Central, Northern and Western Europe is a matter of convention. To avoid ambiguity, we rewrote the sentence as follows: During the Late Cretaceous/Earliest Paleogene, Europe was affected by a compression event. 6. Line 20: you are right; added: transpression at normal faults with high angle to compression. 7. Line 21: added: de Jager 2003 8. Line 24: the main inversion structures in Central Europe (north and east of the Rhenish Massif) show indeed mostly sands and conglomerates and hemipelagic marlstones (also marls are partly composed of clays), but we add: and redeposited carbonates 9. Line 24: This statement is true, because the amount of inversion at the basement uplifts is between 5 and 12 km, in comparison to 500-2000 m

at structures outside this belt. All basement uplifts are arranged in this belt. Important: As this sentence re-appears some lines below, it will be skipped from this place. 10. Line 35 (extended figure): this is a good point, but would be very time-consuming and the figure would be rather complex. Should be an additional paper. Altmark basin (AM) was added to the figure. 11. Line 40: authors added: (Lange et al., Senglaub et al., Danisik et al, von Eynatten et al.) 12. Line 54: added (slickensides, fold axes and fault orientation) 13. Line 58: added: furthermore 14. Line 60: added: on the base of detailed fault analysis, 15. Line 75: foreland is the structural relationship to the inversion structure, marginal trough the resulting structure, (…means we will keep it) 16. Line 79: you are right, changed: …within the Danian. This is in agreement with the results of Deckers and van der Voet (2018) for the timing of inversion in the Roer-Graben and the West-Netherland basin inversion 17. Line 80: better explained: Due to lacking Paleocene deposits and later erosion of both marginal troughs and uplifting structure, these potential secondary marginal troughs are not preserved 18. Line 84: added Central Europe 19. Line 87: changed as proposed 20. We are sure that at least these "pulses" of Late Cretaceous inversion are artefacts, produced by the interaction of continuous tectonics and global sea-level changes. Unconformities develop due to erosion during base-level fall and are covered during base-level rise. These pulses correlate perfectly with late Cretaceous transgressions (see Voigt et al 2004: Late Cretaceous unconformities in the Subhercynian Cretaceous Basin (Germany). The unconformity becomes obvious not until the underlying succession is covered by the next sequence. Therefore, also the interpretation of Betz (1987) of the LSB was in error. 21. Pulsating changes of intraplate-stress patterns are difficult to explain and not really verified by data (explanation follows in the text). We see on all active structures a continuous rotation of the involved succession and no changes in subsidence/sedimentation rates in the basins. Most published data fit perfectly to the observed evolution of unconformities. We published our interpretation (Voigt et al 2004) before, but the implications were not discussed in some following papers about inversion tectonics in the North Sea (probably too strongly focused on their

own targets of investigation). We extended this section. 22. Line 103-105: Since we are concerned with the termination of the Late Cretaceous event, the interpretation of the Cenozoic events is not of key importance for our analysis. Nevertheless, we have added a short paragraph introducing the concept of discrete deformation events with new references as suggested. 23. Line 125: Late Cretaceous 24. Line 126: Late Jurassic-Lower Cretaceous 25. Line 160: fig. 1 26. Line 190: changed: The whole sentence was deleted. As the basins were not interesting for oil exploration, no high quality seismic data exist. 27. Line 289: repeated section was deleted 28. Line 292: Osning thrust was added on Fig. 2 29. Line 309: Jurassic to Lower Cretaceous graben fill 30. Fig 4: Q1 was enlarged 31. Line 212: Betz et al. 1987: We found only one hint to the Turonian slumps at the southern border of the basin in their paper. The mentioned clastic carbonates and iron ores (Upper Campanian) were deposited on top of the inverted basin (Damme syncline). Furthermore, there is no clear evidence for a Laramide phase of inversion, because the first post-inversion deposits are of Rupelian age. There is some minor evidence for post Campanian tectonics (folding, fault with minor displacement, tens of metres) which is either late Cretaceous or early Paleogene in age. 32. Line 369: Fault is already displayed on fig. 4., but was added on fig. 5 33. Line 379: The possible timing of this unconformity spans at least 35 Ma, all of the late Cretaceous and Paleogene phases 34. Line 441: principal stream/principal NW-directed river 35. Line 446: you are right, this is a better expression 36. Line 454: it was changed 37. Line 455: thank you for this precise information, we added it 38. Line 460: the sentence refers to northern Germany 39. Line 470: initially we wanted to concentrate on the German basins, but we will add your advices, because it fits to the overall picture and gives better time-constraints than further to the east. The paper of Best et al 1987 is not the actual state of the art (several papers of Baldschuhn and Kockel in the nineties; the best actual summary is in Littke et al (2008): Dynamics of complex intracontinental basins; Paleocene is only north of the LSB, the Thanetian mirrors the same structural style as the Ypresian and the following Eocene succession, We found no source which confirms the occurrence of Danian on
the inverted LSB 40. Line 509: We will add this 41. Line 528: corrected 42. Line 535: we will include the hint to the Netherlands! 43. Line 547-549: follows in the discussion, at this place we are not able to say anything 44. Line 660 (start of marginal troughs at the borders of the Roer Valley Graben). The Roer Valley Graben and inverted Central Netherlands basin seem to be responsible for the delivery of sand towards the east (Münsterland Basin), starting in the Santonian (not before), according to Luijendijk et al (2011) the exhumation (amount of inversion) of the RVG is weak (between 1000-1250 m); possibly the differentiated subsidence started later. 45. Line 715: Could it be that these phases were a continuous process, and the unconformities were formed due to the fluctuating sea-level? Another explanation would be decoupling along the extensional system Upper Rhine-Graben – Lower Rhine-Graben. There is no evidence for major intra-Cenozoic uplift east of it. 46. Line 720 (fig): comment: It would be nice to compare these with other basins in Western Europe because based on your data, it is difficult to constrain the end of inversion... This is true, the only evidence is the deformation of the Damme syncline (folding and central thrust), which is close to the border of the Netherlands. Possibly one of the Paleogen pulses, but impossible to date. 47. Line 725 (fig text): we added the major unconformities in Germany 48. Line 748: no problem with this, added 49. Line 756: Maastrichtian added

Please also note the supplement to this comment:
https://se.copernicus.org/preprints/se-2020-188/se-2020-188-AC2-supplement.pdf

---

## Author Response (AR2)

**List of modifications**

We have adopted all but very few of the modifications suggested by the editor. Line numbers refer to the pdf file "se-2020-188-comments-to-author". In the list below, "corrected" refers to technical errors, "modified" to adoption of the editor´s preferred formulation, and "rephrased" to more extensive reformulation. For convenience we have highlighted the instances where we don´t follow the editor´s suggestions.

l. 14: clarified

19: modified according to suggestion

20: "Paleogene" deleted

24: corrected, references added

27: modified

30: corrected

35 (Fig 1): Fault pattern is originally not from cited reference but Geological Map of Poland. The compliation shown here is based on Kley & Voigt (2008) as indicated. Other changes made.

p. 2, bottom, location of Fig. 2: Not an overview map but referring to thickness variations in the Lower Saxony Basin first described in chapter 4.

47: modified

74: Unchanged. Since most basins we discuss are closer to the Alps than the Carpathians their inversion is unlikely to have been caused by events in the Carpathians.

77: corrected

79: corrected

83: corrected

86: Rephrased to conform with cited reference

89: corrected

93-99: corrected, 94: rephrased

107: unchanged. Here, we really mean deposition directly on the unconformity surface, not generally above it.

113: unchanged. All references cited thereafter are more recent than those cited before, but not exactly recent anymore (e.g., 2005).

133: rephrased

134 ff. (not marked by editor): rephrased, additional references included

139: rephrased, reference to Fig. 1 added.

146: "in the basins studied" added

157: rephrased

177: modified

182: Fig. 1, changed labels to P and L-S-H, added label NSB, modified caption.

194: corrected

198: corrected

203, 204: corrected

210: rephrased

212: references added

227: corrected

228: Deleted "The"

246: Late Cretaceous marginal troughs of the North-Sea, accompanying the inverted Sole Pit Basin, Broad Fourteens Basin and the Central-Netherlands Basin, the Oldenburg and Münsterland Basins in northern Germany, and the marginal troughs at the Mid-Polish Swell and the Danish Basin were filled with authochthonous and re-deposited fine-grained deposits, marls, hemipelagic limestones, and chalks.

249, 253: Modified

256: Reference to Fig. 1 added

290: added "elastic"

291: modified

302: modified

308: modified

336: corrected

338: Label added to Fig. 3

343: corrected

363: changed: "thicknesses"

397,398: corrected

421: corrected

435: modified

439: modified

452: corrected

458: clarified

474: corrected

480-482: corrected lower case, retained "regardless of", added "or not"

495: Dutch

500, 501: modified

538: modified:  (Cenomanian thickness has not been studied yet in detail)

560: modified

569: Timing is explained in the next few sentences

574: rephrased

575: modified

635: corrected

708: added "major", otherwise unchanged. As basin inversion with no clearly observable reverse reactivation of normal faults does occur, we think the meaning is clear.

760: corrected

764: modified

772: It´s in the following sentence

785: rephrased

796,797: modified

819: modified

904: corrected